# Systematic review and meta-analysis of ex-post evaluations on the effectiveness of carbon pricing

Niklas Döbbeling-Hildebrandt [1,2] ✉, Klaas Miersch [1,3], Tarun M. Khanna [1,4], Marion Bachelet[1], Stephan B. Bruns [5,6,7], Max Callaghan [1], Ottmar Edenhofer[1,3,8], Christian Flachsland[1,9], Piers M. Forster [2], Matthias Kalkuhl [1,10], Nicolas Koch [1,11], William F. Lamb [1,2], Nils Ohlendorf[1,3], Jan Christoph Steckel [1,12] & Jan C. Minx [1,2] ✉

Today, more than 70 carbon pricing schemes have been implemented around the globe, but their contributions to emissions reductions remains a subject of heated debate in science and policy. Here we assess the effectiveness of carbon pricing in reducing emissions using a rigorous, machine-learning assisted systematic review and meta-analysis. Based on 483 effect sizes extracted from 80 causal ex-post evaluations across 21 carbon pricing schemes, we find that introducing a carbon price has yielded immediate and substantial emission reductions for at least 17 of these policies, despite the low level of prices in most instances. Statistically significant emissions reductions range between −5% to −21% across the schemes (−4% to −15% after correcting for publication bias). Our study highlights critical evidence gaps with regard to dozens of unevaluated carbon pricing schemes and the price elasticity of emissions reductions. More rigorous synthesis of carbon pricing and other climate policies is required across a range of outcomes to advance our understanding of "what works" and accelerate learning on climate solutions in science and policy.

Countries are not on track to meet the climate goals they committed to under the Paris Agreement[1,2]. To step up implementation, learning about what policy instruments work in reducing emissions at the necessary speed and scale is critical. But despite more than three decades of experience with carbon pricing and more than 70 implementations of both carbon taxes (37) and cap-and-trade (36) schemes around the world[3] at national, regional and sub-national level, there remains no consensus in science nor policy as to how effective such policies are in reducing greenhouse gas (GHG) emissions.

Proponents have suggested carbon pricing as a key instrument to incentivise GHG emissions reductions on the basis that it would avoid the need for detailed regulatory decisions targeted at specific emission sources[4–8]. However, the effectiveness of carbon pricing is highly dependent on the context and the effect could be higher or lower based on the institutions and infrastructures[9,10]. Critics doubt the

[1]Mercator Research Institute on Global Commons and Climate Change, Berlin, Germany. [2]Priestley International Centre for Climate, School of Earth and Environment, University of Leeds, Leeds, UK. [3]Technische Universität, Berlin, Germany. [4]University of British Columbia, Vancouver, BC, Canada. [5]Centre for Environmental Sciences, Hasselt University, Hasselt, Belgium. [6]International Center for Higher Education Research (INCHER), University of Kassel, Kassel, Germany. [7]Meta-Research Innovation Center at Stanford (METRICS), Stanford University, Stanford, CA, USA. [8]Potsdam Institute for Climate Impact Research, Potsdam, Germany. [9]Hertie School Centre for Sustainability, Berlin, Germany. [10]Faculty of Economics and Social Sciences, University of Potsdam, Potsdam, Germany. [11]Institute of Labor Economics (IZA), Bonn, Germany. [12]Brandenburg University of Technology, Cottbus, Germany. ✉e-mail: doebbeling@mcc-berlin.net; minx@mcc-berlin.net

ability of carbon pricing to unlock the investments required for the development and application of low carbon technologies[11]. There are also concerns about whether policymakers can overcome political barriers and raise carbon prices high enough to deliver emissions reductions at the scale and pace required[11–13].

We aim to systematically review the empirical literature on the effectiveness of carbon pricing policies in reducing GHG emissions. While there are other market based policy instruments, such as fuel taxes, import taxes or value added taxes, we focus here on policies which impose a carbon price across fuels based on their carbon contents. One way to assess the effects of carbon pricing is to evaluate experiences in the real world. A growing scientific literature has provided quantitative evaluations of the effects of different carbon pricing schemes on emissions[14–16]. This evidence is usually provided in the form of quasi-experimental studies which assess the effect of the introduction of the policy (treatment effect). Based on this evidence, our meta-analysis addresses the question: What was the emissions reduction effect of the introduction of a carbon price during the early years of its application? This is different from the question, how emissions respond to gradual changes in existing carbon prices. There exist only very few studies estimating this relationship between the carbon price level and emissions[17–19]. The comprehensive literature on the elasticity of fuel use in response to fuel price changes has been reviewed before in a number of meta-analyses[20–24].

We focus on the growing evidence base on the effectiveness of introducing a carbon price. Reviews of this literature have tended not to employ rigorous systematic review methods such as meta-analysis. A number of reviews describe the literature and summarise the findings of the primary studies but do not attempt a quantitative synthesis of the findings[15,25–27]. Green provides a range of effect sizes reported in the reviewed literature without any formal methodology for their harmonisation and analysis, concluding that the policy has no or only a very small effect on emission reductions (0–2%)[28]. None of the available reviews provide a critical appraisal of the quality of the primary studies considered. Biases of such traditional literature reviews have been widely documented in the literature[29,30]. The lack of comprehensive systematic review evidence on a multitude of policy questions hampers IPCC assessments to learn from implemented climate policies[31–33].

We fill this gap by conducting a systematic review and meta-analysis of the empirical ex-post literature on the effectiveness of carbon pricing, covering 21 enacted carbon tax and cap-and-trade policies around the globe following the guidelines by the Collaboration for Environmental Evidence[34]. We use a machine-learning enhanced approach as proposed by Callaghan and Müller-Hansen[35] to screen 16,748 studies from five different literature databases, identifying 80 relevant ex-post policy assessments. We extract and harmonise estimates of average emissions reductions from the introduction of a carbon price. We conduct a meta-analysis on 483 effect sizes on 21 different carbon pricing schemes and estimate emissions reduction effects. We study the heterogeneity in the reported findings and conduct a critical appraisal as well as a publication bias assessment to analyse the impact of different study design choices on the results. Our methodology is transparent and reproducible, ensuring that our analysis is updatable in the future as new information and experiences with carbon pricing policies are gained around the world[36]. The data and code is publicly available: https://github.com/doebbeling/carbon_pricing_effectiveness.git.

We find consistent evidence that carbon pricing policies have caused emissions reductions. Statistically significant emissions reductions are found for 17 of the reviewed carbon pricing policies, with immediate and sustained reductions of between −5% to −21% (−4% to −15% when correcting for publication bias). Our heterogeneity analysis suggests that differences in estimates from the studies are driven by the policy design and context in which carbon pricing is implemented, while often discussed factors like cross-country differences in carbon prices, sectoral coverage, and the design of the policy as a tax or trading scheme do not capture the identified heterogeneity in effect sizes.

## Results

### Evidence base − larger and more diverse than previously suggested

With the help of our machine-learning assisted approach, we identify 80 quantitative ex-post evaluations across 21 carbon pricing schemes around the globe (see Methods). Previous reviews covered a maximum of 35 research articles on the emissions reduction effect of carbon pricing policies[15,25,26,28].

As shown in Table 1, the carbon pricing schemes covered here are very diverse and differ in terms of their specific policy design, scope, and policy context. For example, some of the schemes are targeted at large scale emitters in the industry and energy sectors, while others focus on households via home energy use and the transport sector. In the European Union, some sectors are regulated with a carbon tax while others are covered by the European wide emission trading scheme. We also observe substantial differences in carbon price levels of the covered schemes. All of these differences may give rise to considerable variations in emissions reductions achieved.

Beyond these differences in policy design, carbon price levels, and regional contexts, all considered policy experiences speak to the question whether carbon pricing is or is not effective in reducing GHG emissions. A systematic assessment and comparison of the outcomes of these policies can inform policymakers and future research by synthesising the available evidence.

The number of available ex-post evaluations on the effectiveness of carbon pricing differs substantially across schemes. Prior reviews suggested a bias towards evaluating schemes in Europe and North America[15,26,28], however here we find that the vast majority of the available ex-post evidence on the effectiveness of carbon pricing assess the pilot emission trading schemes in China − 35 of the 80 articles. There are 13 studies on the European emissions trading scheme (EU ETS), seven on British Columbia and five on the Regional Greenhouse Gas Initiative (RGGI) in the United States. The remaining schemes are evaluated by a single or very few studies.

Our systematic review also reveals some fundamental evidence gaps in the literature. Despite the broad set of bibliographic databases searched, we found evidence only for 20 out of 73 carbon pricing policies in place in 2023[3] and for the Australian carbon tax, which was repealed two years after its implementation. For some, more recently implemented, policies this may be explained by the time needed for sufficient data to become available, be assessed, and the results published. But even of the 38 carbon pricing schemes already implemented by 2015, for 18 of these we could not find a single study on effectiveness, despite the broad set of bibliographic databases searched (see Supplementary Information). There is also little evidence on the effectiveness of carbon pricing relative to the level of the carbon price (carbon price elasticity). We identify only nine price elasticity studies, providing too few effect sizes for meta-analysing these separately.

### Average emissions reductions across carbon pricing schemes

In order to provide a meaningful and transparent synthesis of the available quantitative evidence, we harmonise the effect sizes extracted from the individual studies to a common treatment effect metric following standard systematic review guidance[34]. This treatment effect is expressed as the percentage difference between the counterfactual emissions without carbon pricing and observed emissions after the introduction of a carbon price. It assumes emissions reductions to take place at the time of the introduction of the policy and to persist throughout the observation period as a constant difference to

**Table 1 | Carbon pricing policies**

| Policy | Jurisdiction | Introduction | Sector coverage | Emission coverage | Mean price | Studies | Effect sizes |
|---|---|---|---|---|---|---|---|
| Chinese pilot ETS | | | | | | 46 | 179 |
| o/w Hubei pilot ETS | Hubei, China | 2014 | industry | 27% | $3 | 4 | 13 |
| o/w Beijing pilot ETS | Beijing, China | 2013 | industry, power, transport and buildings | 24% | $8 | 2 | 3 |
| o/w Shanghai pilot ETS | Shanghai, China | 2013 | industry, buildings, transport | 36% | $4 | 2 | 3 |
| o/w Guangdong pilot ETS | Guangdong, China | 2013 | industry, aviation | 40% | $5 | 2 | 2 |
| o/w Shenzhen pilot ETS | Shenzhen, China | 2013 | industry, power, buildings, transport | 30% | $7 | 1 | 2 |
| o/w Tianjin pilot ETS | Tianjin, China | 2013 | industry, buildings | 35% | $4 | 2 | 2 |
| EU ETS | 30 European countries | 2005 | power, manufacturing industry, aviation | 38% | $20 | 13 | 77 |
| Swedish carbon tax | Sweden | 1991 | transport, buildings | 40% | $103 | 2 | 77 |
| BC carbon tax | British Columbia, Canada | 2008 | industry, power, transport and buildings | 70% | $18 | 7 | 39 |
| Saitama ETS | Saitama, Japan | 2011 | industry, power, buildings | 17% | $108 | 3 | 20 |
| Tokyo ETS | Tokyo, Japan | 2010 | industry, power, buildings | 20% | $106 | 4 | 14 |
| Quebec ETS | Quebec, Canada | 2013 | industry, power, transport and buildings | 77% | $9 | 2 | 10 |
| RGGI | 11 northeastern US states | 2009 | power | 14% | $3 | 8 | 10 |
| UK carbon price support | United Kingdom | 2013 | power | 24% | $22 | 4 | 10 |
| Finnish carbon tax | Finland | 1990 | industry, transport, buildings | 36% | $6 | 2 | 8 |
| Swiss ETS | Switzerland | 2008 | industry, power | 11% | $18 | 1 | 5 |
| Australian carbon tax | Australia | 2012* | industry, power | 60% | $24 | 1 | 2 |
| California CaT | California, USA | 2012 | industry, power, transport, buildings | 74% | $12 | 2 | 2 |
| Korea ETS | Korea | 2015 | industry, power, buildings, domestic aviation, public sector, waste sector | 74% | $15 | 2 | 2 |
| Cross-country | | | | | | 4 | 18 |
| Total | | | | | | 101 | 483 |

All information on the carbon pricing schemes was retrieved from the World Bank[3], except for the price data for the EU ETS, which is retrieved from ICAP[53]. The information for the sector coverage was simplified. For more detailed information on the coverage, including covered or exempted subsectors, the reader is referred to the World Bank data. *Cross-country* studies analyse countries with and without carbon pricing, not focusing on a specific carbon pricing scheme. The effects of the eight Chinese pilot ETS schemes are often analysed collectively in a single study, while some studies focus on individual schemes. We only list pilots that have been studied individually. The Australian carbon tax was revoked in 2014. Mean prices are unweighted average prices in constant 2010 US$ during the period analysed by the studies in our sample. "Emission coverage" is the share of a jurisdictions emissions covered by the carbon price in 2022. The number of studies exceeds the number of reviewed articles, as some articles include more than one relevant study using disparate datasets.

counterfactual emissions. Most studies directly provide treatment effects, which we standardise to represent a percentage change in emissions levels. Effect sizes provided as price elasticity are interpreted at the mean carbon price (see Methods). Overall, we harmonise 483 effect sizes from 80 reviewed articles, covering 21 carbon pricing schemes that provide the starting point for our quantitative synthesis.

Our results show that carbon pricing effectively reduces greenhouse gas emissions. We use multilevel random and mixed effects models to account for dependencies among effect sizes in our sample and estimate the average treatment effects. The mixed effects model includes dummy variables for each of the included carbon pricing schemes to estimate the effectiveness for each of the schemes. As depicted in Panel a of Fig. 1, emissions reduction effects are observed consistently across schemes with considerable variation in magnitude. For 17 of the carbon pricing schemes we find statistically significant average reduction effects from the introduction of a carbon price. The estimated reduction effects range from about −21% to about −5%. Across carbon pricing schemes, we find that on average the policy has reduced emissions by −10.4% [95% CI = (−11.9%, −8.9%)]. This effect is both substantial and highly statistically significant.

The reviewed literature provides large differences in the amount and quality of evidence for individual schemes. Focusing on those with the largest evidence base, we find an average treatment effect for the eight Chinese ETS pilots of −13.1% [95% CI = (−15.2%, −11.1%)], which is higher than the −10.4% average treatment effect across the schemes. The EU ETS and the British Columbia carbon tax both have estimated emission reduction effects below the overall average treatment effect. These are estimated at −7.3% [95% CI = (−10.5%, −4.0%)] and −5.4% [95% CI = (−9.6%, −1.2%)]. Reduction effects smaller than −5% are only reported in three instances with severe problems in study design exposing estimates to a high risk of bias (Korean ETS, Australian carbon tax, Swiss ETS).

**Critical appraisal and publication bias**

The average treatment effects presented in the previous section were based on all reviewed studies. However, the quality of the primary studies is not uniform and some are subject to biases in the study design. Additionally, the average treatment effect might be subject to publication bias. Therefore we re-estimate the treatment effects by adjusting for potential quality issues and publication bias, adopting transparent and reproducible criteria.

We critically appraise each primary study, to identify potential biases in the study design. These biases often arise from the unreasonable selection of a control group used in a quasi-experimental design; from inadequately controlling for confounding factors like the introduction of other relevant policies; or from statistical

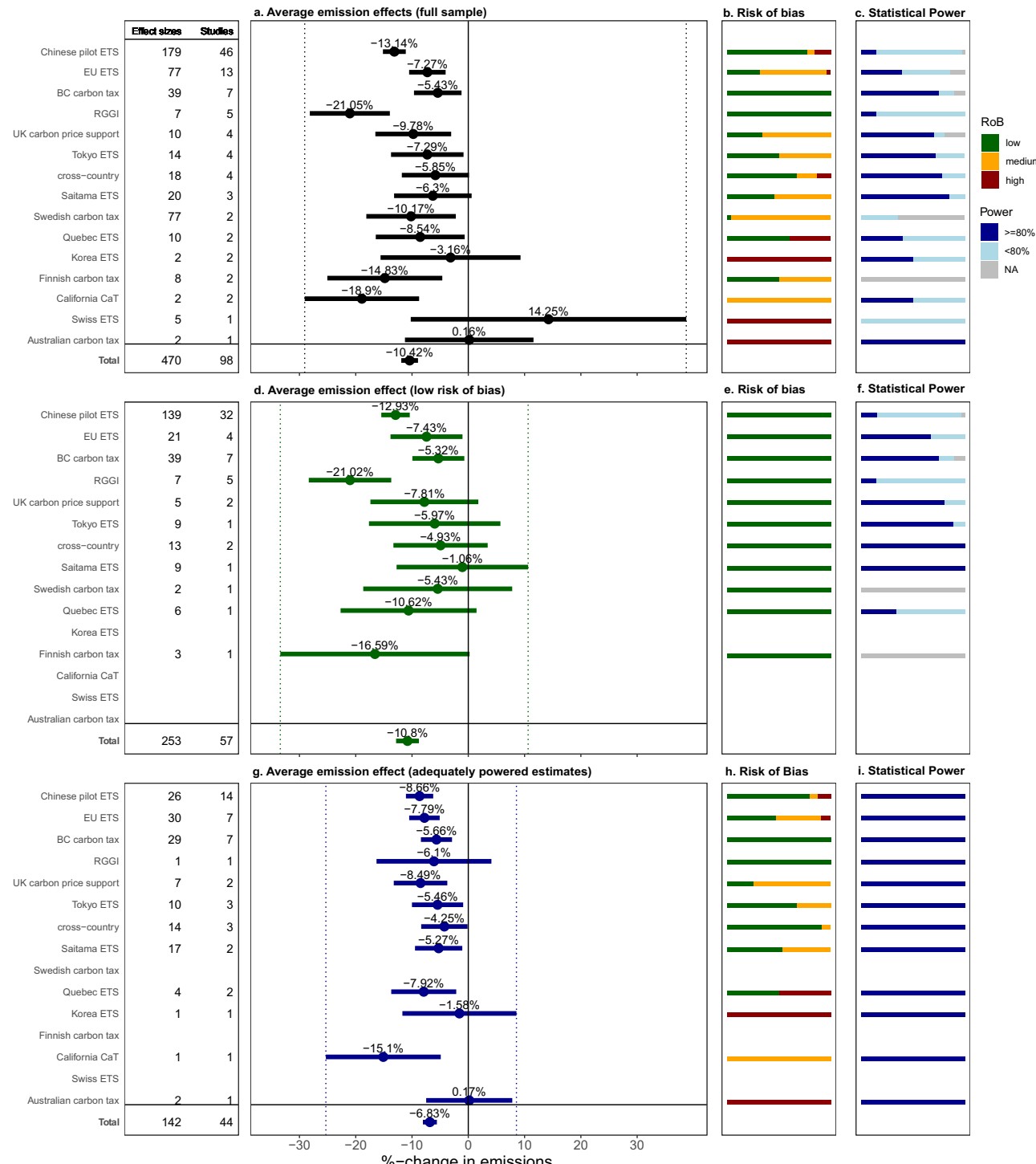

**Fig. 1 | Average emissions changes by scheme.** Panel (**a**, **d**, **g**) present weighted mean effect sizes together with their 95% confidence intervals based on multilevel random and mixed effects models and represent the effect of the policy observed in each period after its introduction in comparison to the counterfactual emissions without the policy. The estimates are ordered according to the number of studies they comprise (depicted on the left). The average treatment effect for the Chinese ETS pilots comprises the effects of all eight regional pilot schemes. *Cross-country* collects the evidence from studies assessing countries with and without carbon pricing, not focusing on a specific carbon pricing scheme. Panel (**a**, **d**, **g**) comprise, respectively *n* = 470, *n* = 253, and *n* = 142 effect sizes clustered on the study level. Panels (**b**, **e**, **h**) show the distribution of assigned risks of bias (RoB). Panel (**c**, **f**, **i**) show the distribution of statistical power. Power above 80% is considered adequate. For synthetic control designs no statistical power was derived, thus presented as "NA".

specifications that do not allow to single out the policy effect. The assessment criteria for the critical appraisal are set out in the methods section and the Supplementary Information. 46% of the reviewed studies are assessed to have a medium or high risk of bias. When we remove studies with medium or high risk of bias from the sample, the

average treatment effects for some of the schemes are adjusted by up to 5 percentage points, while the estimation uncertainty increases due to the reduction of considered primary estimates (see Fig. 1, Panel d). The identified biases, however, do not systematically impact the estimated treatment effects in either direction. The average treatment

effect across policies is practically unchanged when removing studies with medium or high risk of bias.

Secondly, we adjust the average treatment effect for the influence of publication bias. Publication bias could arise from a tendency in the literature towards only publishing statistically significant effects[37–40]. A precision effect test[41,42] confirms the presence of publication bias in the set of studies reviewed here (see Supplementary Information). As suggested in the literature, we correct for publication bias by estimating average effects for a subsample of effect sizes with adequate statistical power (see Methods)[38], which applies to about 30% of the reviewed effect sizes. This subsample analysis adjusts most of the scheme-wise average treatment effects towards lower estimated emissions reductions (see Fig. 1, Panel g), ranging from −15% to −4%. Across the schemes, the average treatment effect is reduced to −6.8% [95% CI = (−8.1%, −5.6%)]. Despite these adjustments, the publication bias corrected estimates support the overall finding that carbon pricing policies cause significant reductions in in GHG emissions.

Studies with a high risk of bias and low power are not uniformly distributed across schemes. Some schemes are evaluated only by a few biased studies, resulting in very high or low average treatment effects. For example, when considering all available evidence, the carbon pricing schemes in South Korea, Switzerland, and Australia are estimated to have the lowest negative or even positive average treatment effects. These estimates are based entirely on studies with a high risk of bias and are no longer considered when re-estimating the treatment effects based on low risk of bias studies (see Fig. 1d). The two carbon pricing policies from the United States (California CaT, RGGI), which show the largest negative average treatment effect when considering all available studies, show lower average treatment effects after the adjustment for publication bias (see Fig. 1g). For other schemes, like the EU ETS and British Columbia's carbon tax, there is no substantial change in the average treatment effect when studies with high risk of bias are excluded.

## Explaining heterogeneity in effect sizes

There is considerable variation in the effect sizes reported by primary studies included in this review. This could arise from heterogeneity in the design of the carbon pricing policies or from heterogeneity in the design of the primary studies. The carbon pricing literature mainly discusses three policy design factors that could potentially explain differences in the effectiveness of the policy. First, there are debates whether carbon prices are better applied as carbon taxes or as emission trading schemes[5,43–47]. Secondly, it is argued that the policy causes different reduction rates in different sectors[48–50]. And thirdly, the level of the carbon price can be expected to play a decisive role for the magnitude of the emission reductions[5,51,52]. We assess whether, and to what extent, such factors are able to explain differences in the treatment effects reported. We test which factors are most relevant to explain the reported emissions reductions by using scheme and study characteristics as explanatory variables in meta-regressions.

As we are confronted with a large number of potentially relevant explanatory variables, we use Bayesian model averaging (BMA) to assess the heterogeneity in the estimated effect sizes reported by the different studies. BMA is particularly suitable for meta-analysis as it allows for running a large number of meta-regressions with different possible combinations of explanatory variables and does not require selecting one individual specification (see Methods). We include explanatory variables for the three policy design factors provided above: price level, sector coverage, and a variable differentiating between carbon taxes and cap-and-trade schemes. In addition we add dummy variables for each of the carbon pricing schemes, capturing the remaining policy design and contextual factors of each policy scheme. Additionally, we test whether studies assessing longer periods after the policy implementation find higher or lower treatment effects. To assess the impact of methodological choices made in the studies,

## Table 2 | Heterogeneity assessment using Bayesian model averaging

|  | PIP | Post mean | Post SD |
|---|---|---|---|
| **RGGI** | **1.00** | **−28.45** | **5.09** |
| **Chinese_pilot_ETS** | **0.99** | **−9.76** | **2.23** |
| **Swiss_ETS** | **0.80** | **14.35** | **8.93** |
| **Data_City** | **0.78** | **11.39** | **7.63** |
| **duration** | **0.76** | **−0.64** | **0.46** |
| synthetic_control | 0.42 | 2.87 | 3.87 |
| tax | 0.41 | −3.11 | 4.24 |
| BC_carbon_tax | 0.38 | 3.90 | 5.65 |
| Swedish_carbon_tax | 0.36 | −3.05 | 4.65 |
| Coal | 0.32 | −2.58 | 4.26 |
| Less_Bias | 0.30 | 1.16 | 2.00 |
| Finnish_carbon_tax | 0.25 | −2.89 | 5.70 |
| TransLevelLevel | 0.19 | −0.70 | 1.67 |
| Data_Region | 0.12 | −0.40 | 1.30 |
| log_carbon_price | 0.09 | 0.15 | 0.62 |
| Data_Sector | 0.09 | 0.25 | 1.02 |
| Gas | 0.08 | −0.44 | 1.88 |
| other_schemes | 0.05 | −0.31 | 2.00 |
| Tokyo_ETS | 0.04 | 0.15 | 1.03 |
| industrial_sectors | 0.04 | −0.04 | 0.77 |
| Data_Firm | 0.04 | 0.07 | 0.54 |
| Data_Plant | 0.03 | 0.04 | 0.47 |
| DVTotal | 0.03 | 0.03 | 0.51 |
| Saitama_ETS | 0.03 | 0.05 | 0.65 |
| SE_percent | 0.03 | −0.00 | 0.00 |
| Gasoline | 0.03 | −0.02 | 0.62 |
| Quebec_ETS | 0.03 | −0.06 | 0.82 |
| Data_Month | 0.03 | −0.02 | 0.52 |
| Data_Year | 0.03 | 0.01 | 0.41 |
| Data_Airline | 0.03 | −0.00 | 0.75 |
| (Intercept) | 1.00 | −5.99 |  |

The table provides the results of meta-regressions using Bayesian model averaging. The dependent variable for each of the meta-regression models is the percentage change in emissions. The posterior inclusion probability (PIP) indicates the relevance of each variable. Variables with PIP≥0.5 are considered relevant for explaining the heterogeneity in carbon emissions reductions reported across primary studies. Post Mean and Post SD represent the mean and standard deviation of the posterior distribution for a respective explanatory variable. Five variables have PIP≥0.5 and are considered relevant (marked in bold): the dummy variables for *RGGI*, *Chinese pilot ETS*, *Swiss ETS*, *Data_City*, and *duration*. The dummy variables represent the geographic location in which the policy was implemented, with the reference location being EU ETS. *Data_City* captures whether primary studies used city level data versus country level data. The variable *duration* captures the number of years for which data on the scheme was collected after the policy was implemented. Definitions of the other explanatory variables are provided in the Supplementary Information.

we study a set of variables including the type of study design, estimation method, and data used in the primary studies.

The results from the BMA are provided in Table 2 and Fig. 2. The posterior inclusion probability (PIP) indicates the relevance of each variable. Commonly, variables with a PIP above 0.5 are interpreted to be relevant explanatory factors, while variables with lower PIPs are unable to capture the observed heterogeneity. The table furthermore provides the posterior mean and standard deviation of the estimated effect averaged across all meta-regressions that include the respective variable.

Variation in carbon prices, the sectoral coverage of schemes, and choice of carbon tax vs. cap-and-trade do not seem to be important variables in explaining the observed heterogeneity in emissions reductions (PIP < 0.5). Instead the dummy variables for the place where

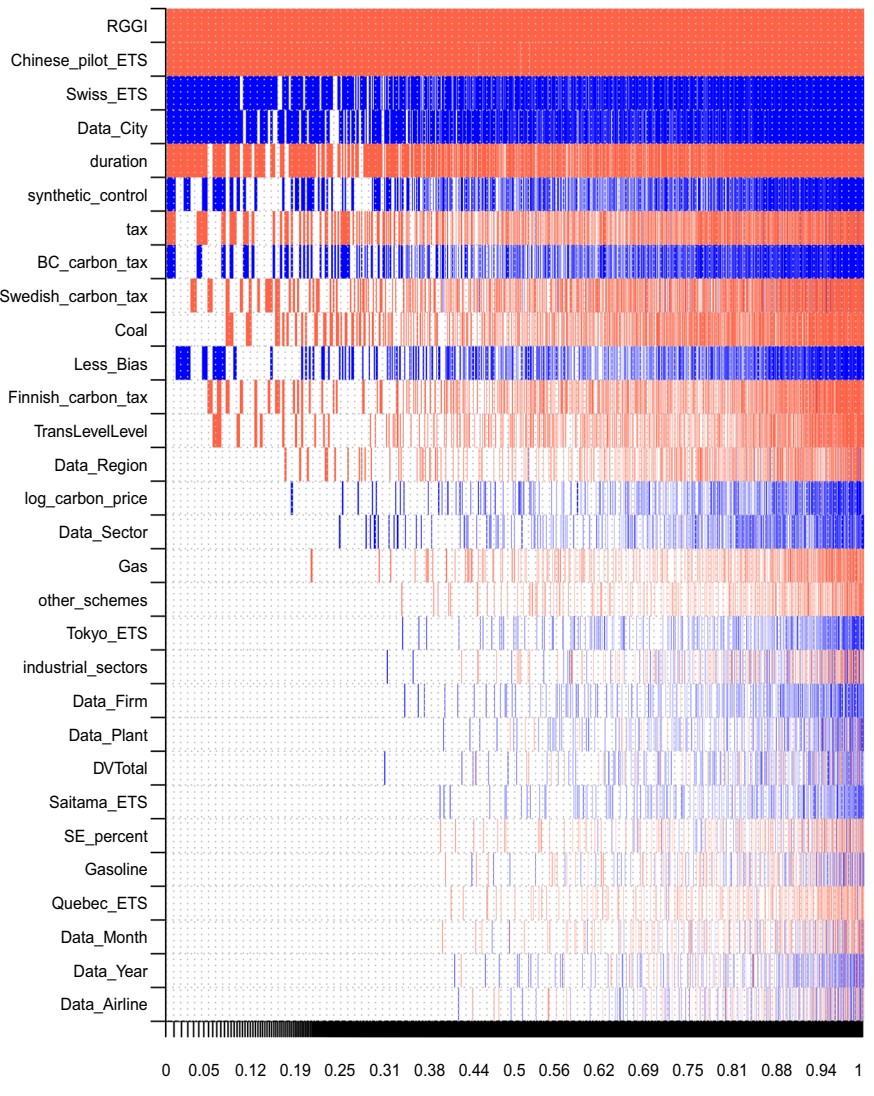

**Fig. 2 | Heterogeneity assessment using Bayesian model averaging.** The columns in the figure depict the best 26,435 estimated meta-regressions, with each column showing the outcome of one estimated meta-regression model. The dependent variable for each of the meta-regression models is the percentage change in emissions. The possible explanatory variables are depicted in the rows (ordered by their PIP in descending order) and the explanatory variables included in a respective meta-regression model of the column is indicated by the colours. Red colour indicates the variable was included with a negative sign (larger emission reductions). Blue colour indicates a positive sign (smaller emission reductions). No colour indicates that the variable was not included in the meta-regression model represented by that column. The horizontal axis indicates the cumulative posterior model probabilities across all models. The models are ranked by their posterior model probability with the model on the left accounting for the largest posterior model probability. The definitions of the explanatory variables are provided in the Supplementary Information.

the schemes are applied do a better job in explaining this heterogeneity than the variables that capture specific design characteristics. The variables for the RGGI and the Chinese ETS pilots have a larger reduction effect on emissions than the EU ETS, which is set as the reference category. The Swiss ETS is estimated to have less of a reduction effect compared to the benchmark. Alternative specifications of the BMA, provided in the Supplementary Information, also estimate a larger reduction effect for the Swedish carbon tax compared to the benchmark. The directions of these coefficients are in line with the average treatment effects presented in Fig. 1, for the respective geographies.

If we remove the dummy variables for the schemes, the size of the carbon price becomes an important variable in the BMA to explain the heterogeneity in emission reductions with a PIP close to 1 (see Supplementary Information). However, in the absence of the scheme dummies the effect of the price variable is likely to be confounded as

the scheme dummies account for any omitted context variable that does not vary within a scheme. The high correlation of 0.96 between the scheme dummies and the price variable indicates that the price variable captures the heterogeneity between schemes. In fact, the price coefficient is estimated with a positive sign in the BMA specification without the scheme dummies, implying that lower emissions reductions are achieved with higher carbon prices. The counter-intuitive direction of the price effect indicates a misspecification of the model when the scheme dummies are excluded. Below we discuss possible causes for this inverse relationship between the price and the reduction effect in our data. The effect of carbon prices on emissions reductions is better identified by adding scheme dummies to focus on the variation of prices within each scheme. However, the largest share of the variation in our carbon price variable comes from variation between the schemes (91%) and only 9% from within scheme variation. This is not a limitation of our dataset. Indeed, carbon prices tend to

vary strongly across countries based on the design and coverage of the scheme. But for individual schemes prices have historically been stagnant (EU ETS till recently, RGGI, Chinese ETS pilots) or increases relatively modest (BC carbon tax)[53] and the effect size estimates evaluated here provide limited time frequency. We suspect that due to this low variation, our sample has insufficient power to identify carbon prices as a relevant factor in explaining emissions reductions.

Studies assessing the effectiveness of carbon pricing over longer time periods find larger emission reductions. The coefficient for the variable *duration* has a PIP of 0.76 and is estimated with a negative sign for all regression specifications it is included in. Testing for the spatial and temporal granularity of the data suggests that only the use of city level data compared to the country level explains some of the heterogeneity in reported effect sizes. Methodological differences in the reviewed studies only have a minor influence on effect sizes. These are discussed in further detail, in the Supplementary Information.

In line with the previous section, we also include the risk of bias variable and the standard error, capturing the publication bias. They are both not detected to be most relevant to explain the heterogeneity.

## Discussion

In this first quantitative meta-analysis of carbon pricing evaluations, we find robust evidence that existing carbon pricing schemes that have been evaluated to date are effective in reducing GHG emissions. Our machine-learning enhanced approach to study identification finds more than twice as many ex-post evaluations than existing reviews[15,25,26,28], studying the effectiveness of 21 carbon pricing policies. Our meta-analysis finds that at least 17 of these policies have caused significant emissions reductions ranging from −5% to −21%. These are substantially larger than the 0% to −2% suggested in the recent and widely cited review by Green[28], which lacks a clear and transparent methodology to synthesise the literature[29], not allowing us to formally compare our results. Our finding is robust to biases from poor study designs as well as publication bias. Correcting for the latter adjusts the range of observed emissions reductions to −4% to −15% across carbon pricing schemes.

The synthesis of research findings across carbon pricing schemes provides comprehensive and consistent evidence of its effectiveness, despite the heterogeneity of policy designs and regional contexts. Compared to the recent assessment report by the IPCC, which provides a quantification of achieved reductions only for the EU ETS[54], our systematic review adds synthesised emission reduction estimates for more than a dozen carbon pricing schemes. We provide these estimates together with uncertainty ranges and a transparent assessment of study quality and highlight the presence of substantial variation in emissions reductions achieved across the schemes in our sample, ranging from −5% for the carbon tax in British Columbia to −21% for the RGGI. We conduct an early application of Bayesian model averaging for meta-regressions on our dataset of 483 effect sizes to disentangle which factors explain these differences. The findings suggest that the individual context and policy design of the schemes best explain the heterogeneity in achieved emissions reductions. These are the most relevant explanatory factors despite controlling for broader policy design features like the sectoral coverage or the design as carbon tax or carbon trading scheme as well as for study design features of the primary studies.

Our heterogeneity analysis does not identify a relationship between the price level and the achieved emissions reductions, i.e. the size of the emissions reductions observed across schemes from the introduction of a carbon price cannot be explained well by the carbon price level. This is not surprising as marginal abatement costs may differ widely as, for example, prominently acknowledged in the literature on linking carbon pricing schemes[55,56]. It is further different from the expectation that higher carbon prices lead to larger emissions

reductions within a carbon pricing scheme as commonly found in available assessments of fuel price elasticities[24,57,58]. In line with this argument, we find that the relationship between carbon price levels and emissions reductions in our meta-analytic framework is dominated by the across-scheme variation in prices, which accounts for 91% of the variation in our dataset while the variation within schemes only accounts for 9%. The interpretation for not finding a clear relationship should thus rather be that when implementing a carbon price in two countries with different country contexts, the country with the higher carbon price would not necessarily experience the higher emissions reductions.

This can be observed, for instance, when looking at the cases of China, the EU, and British Columbia. The reviewed literature finds larger emissions reduction effects for the pilot emission trading schemes in China (−13.1%) than for the EU ETS (−7.3%) and the carbon tax in British Columbia (−5.4%), despite the very low carbon prices of the Chinese schemes. The average prices of the eight Chinese pilot schemes are all below US$ 8 during the study period, while the average prices for the EU and British Columbia are at US$ 20 and US$ 18, respectively. This is likely a result of lower abatement costs in China[59] together with differences in the policy contexts of the countries. The effectiveness is certainly influenced by other policies in place. In China indirect carbon prices are lower than in the EU countries and Canada[60], allowing for a higher marginal effect of the implementation of the ETS pilots in China. Non-pricing instruments also diverge across countries. In addition, the implementation of a carbon price (even with a low price) can have a signalling effect towards the emitters, underlining the commitment of the government towards climate mitigation. Evidence for the Guangdong province suggests that signalling has significantly contributed to the achieved emissions reductions in the context of the introduction of the ETS pilots in China[61]. Another example highlighting the relevance of the context of the policy implementation is the case of the RGGI. The policy implementation coincides with the shale gas boom, which drastically reduced the prices of natural gas in the USA and started around the same time as the RGGI was implemented. In face of these general price dynamics in the US energy sector, RGGI participating states reduced their emissions considerably stronger compared to non-regulated states[62,63], while the carbon price was only US$ 3 on average.

Even if across schemes the price level of the carbon price, is not found to be the relevant driver of the emissions reductions achieved with the introduction of the policy, within a scheme the effectiveness is expected to increase with increasing prices. This is well studied for other changes in fuel prices, which are found to substantially reduce its consumption[57,58]. That literature studies all possible price changes on a single fuel, while the here assessed literature on carbon prices studies the effect of a single policy instrument across all fuels. It is thus a complementary but distinct body of evidence. Meta-analyses estimate a reduction of fuel consumption between 0.31% and 0.85% in the long run for a 1% increase in the fuel price[20–24].

Within the literature evaluating the policy effectiveness we identified only nine primary studies estimating semi-elasticities of carbon prices. Four are using the stepwise introduction of the carbon tax in British Columbia to estimate elasticities for the transport and buildings sectors[17,18,64,65], while one is conducted respectively for RGGI[63] and EU ETS[19]. In addition, some studies estimate elasticities across countries and carbon pricing schemes[15,66,67]. These studies support what was already known from studies on the price elasticity of fuel consumption[20–24,57,58]: increasing prices reduce fuel use and emissions. Hence, as carbon prices further rise after the introduction additional emissions reductions are achieved. Interestingly, some studies suggest that an increase in the carbon tax leads to larger emissions reductions than an increase of the same size in the market price of the fuel[17,18,64,65]. It will thus be a relevant avenue for future research to understand

whether it is a generalisable finding that price elasticities are higher for policy induced price changes compared to market price changes of fossil fuels. Such research could draw on the comprehensive evidence from the fuel price literature.

Our meta-regression results suggest that the policy effectiveness of carbon pricing policies increases with time. Studies covering longer time periods after the introduction of the carbon price report larger emissions reduction effects compared to assessments for shorter time periods. While this finding should be treated with caution, as most of the primary studies assume constant treatment effects for their estimations, it hints towards increasing emissions reductions in the years following the policy introduction. The assumption of constant treatment effects reflects not only methodological considerations of the primary studies, but is also based on the expectation that as long as the carbon price of the implemented policy is unchanged, the emission reduction effect should not intensify. The finding of our meta-regression to some extend counters that assumption. An increasing policy effectiveness could be a result of steady adjustment processes, enforced by innovation and investments into cleaner production and infrastructure. Additionally, the literature reviewed here provides some evidence that an increasing policy stringency has also played its role in strengthening the effectiveness of the policy. Increases in the carbon prices led to additional emissions reductions in Sweden[68] and the United Kingdom[69]. Similar effects are found for the EU ETS, where the effectiveness increases with the increasing stringency from phases I, II, and III[70–73].

While the harmonisation and synthesis of the emissions reduction effects provides an overview of the policy effectiveness across a large number of policy schemes, it raises a number of policy relevant research questions, which cannot be answered with our purely quantitative, meta-econometric approach – which is inherently dependent on the available evidence base. These limitations could be addressed using promising and widely unexplored mixed method review designs such as realist synthesis[74,75] which systematically combine quantitative and qualitative information to better understand why particular policy designs work, under what conditions, and why. Some research gaps, however, need to be filled by further primary research. First, there are more than 50 carbon pricing schemes that have not yet been evaluated for their emission reduction effect, despite some of them being enacted for more than ten years (see Supplementary Information). Others have still been studied insufficiently or only poorly. Second, we lack ex-post evidence of higher carbon prices. There are currently less than ten studies assessing emissions reductions in schemes with mean carbon prices higher than US$ 30 across the observation period. As policy ambitions are raised over time, there is an opportunity to strengthen that evidence base. Thirdly, this systematic review highlights substantial challenges with the quality of available primary evidence. Only about half of the studies assessed here follow rigorous study designs with a low risk of bias and only 30% of the studies are adequately powered. While some of this might be related to a lack of access to adequate data for the most rigorous research designs, high quality primary research is essential to understand the effectiveness of climate policies[76]. The multitude of supplementary or conflicting policies as well as other confounding factors pose a challenge to the clear identification of the causal effects of a specific policy[77]. Novel methods of reverse causality are a promising avenue to address this challenge[78].

The effectiveness is just one dimension of policy outcome relevant to the selection of the best policy measures. Systematic assessments of the ex-post climate policy literature on a multitude of policy outcomes and different climate policy options could be the basis for accelerated learning on climate policies and considerably improve upcoming IPCC assessments. Unless we raise our standards and do this work, policy makers and society will remain in the dark as to the most promising pathways towards addressing the climate crisis.

## Methods

The systematic review broadly follows the guidance for systematic reviews by the Collaboration for Environmental Evidence[34], extended by a machine- learning assisted identification of relevant studies. A description of our methods has been published as a review protocol on OSF Registries in advance[79].

### Literature search

We search the bibliographic databases Web of Science, Scopus, JSTOR, RePEc and the web-based academic search engine Google Scholar using a broad search string which comprises a large set of carbon pricing synonyms and indicator words for quantitative ex-post study designs. The full query can be found in the protocol[79]. After the removal of duplicates the search, conducted in the second week of March 2022, returned a set of 16,748 articles (see Fig. 3).

We screened these articles for their eligibility in two stages. First, we screened them at the title and abstract level using the NACSOS software[80] followed by a screening at full text level. Studies are included if they infer a causal relationship between carbon pricing and the emission development. Eligible studies analyse effects on emission levels or emission levels per capita. Studies were excluded if they assess the effect on emission intensity or emission productivity, i.e. the effect on emissions relative to output. The included policy measures are restricted to explicit carbon taxes and cap-and-trade schemes. Studies on implicit carbon taxes and carbon offsetting mechanisms are excluded. We only include studies published in English language.

The screening at the title and abstract level was simplified by an active learning algorithm, using support vector machines to rank the studies in the order of relevance. We stopped screening when we were 90% confident that we had identified at least 90% of the articles relevant to our systematic review, based on the conservative stopping criterion provided by Callaghan and Müller-Hansen[35]. This reduced the amount of manually screened documents by 77%. All articles included after the title and abstract screening were screened at full text, without any further application of machine-learning algorithms. Figure 3 depicts the articles included and excluded at each screening stage.

### Data extraction and critical appraisal

From the included studies we extract the effect size information, including the estimated effect size and direction of the effect, the uncertainty measure, provided as standard error, t statistic, confidence interval, $p$ value, or the indicated significance level, as well as the provided mean emissions and, for price elasticity studies, the mean carbon price. We also capture information on the studied carbon pricing scheme, time of the intervention, study period, emission coverage (sectors, fuels, gases), study design, and estimation method.

We developed criteria for a critical appraisal, by adapting the ROBINS-I assessment criteria[81] to the specific nature of the research studies at hand. First, while the treatment (i.e. the policy application) in the reviewed studies is independent of the conducted research, the study design should cover a representative sample and suitable data. The control group needs to have high similarity with the treatment group, based on demographic, economic, and institutional proximity and similarity in pre-treatment emissions pathways. Statistical methods such as matching or synthetic control methods can increase the comparability of the control group with the treatment group. Second, the study design must control for confounding factors that are expected to influence the emissions of the study objects. For some studies we identify further risks of bias in the set-up of the statistical methods, which are also recorded.

All extracted data is made publicly available (see Data Availability).

### Standardising effect sizes

We standardise the extracted effect sizes, based on the heterogeneous study designs and estimation methods, into a common metric. The

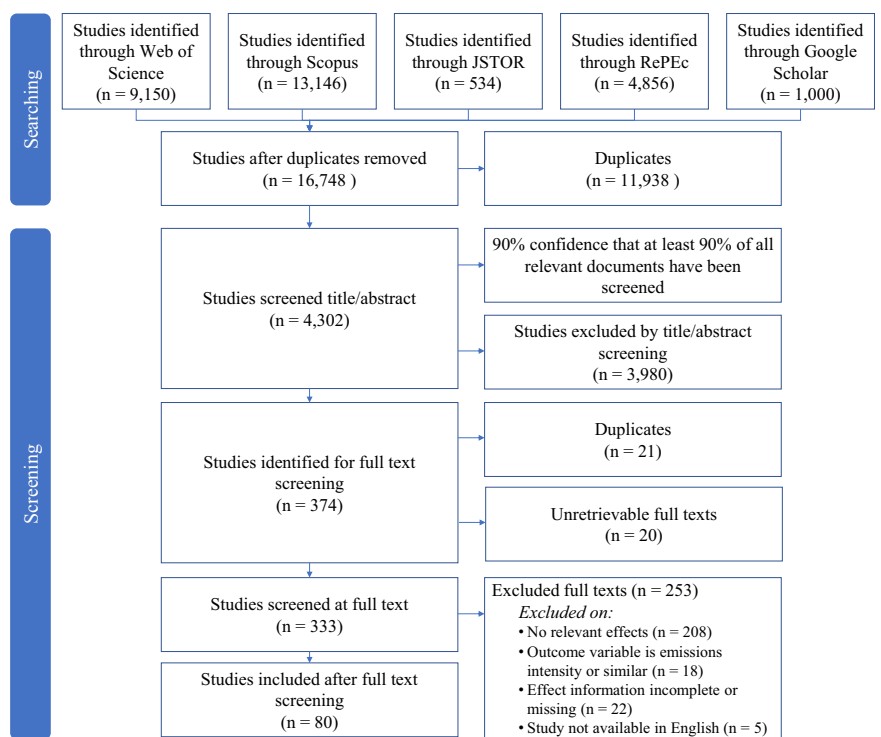

**Fig. 3 | Flow diagram of the literature search and screening process.** Adapted from the ROSES flow diagram for systematic reviews[97].

largest part of the primary literature estimates treatment effects using quasi-experimental study designs (difference-in-differences or regression discontinuity in time). A few studies estimate the treatment effect by comparing the emission levels between countries with and without carbon pricing without any quasi-experimental design (termed *cross-country* studies in this review). Some studies estimate a carbon price elasticity, i.e. the effect of a marginal change in the carbon price on emissions. All effect sizes are transformed to treatment effects measured as a percentage difference between the counterfactual emissions without the policy and the observed emissions with the policy in place. Effect sizes expressed in tons of $CO_2$ are standardised using the mean emissions given in the study, while effect sizes from log-level regression specifications are standardised using exponential transformation. Effect sizes from price elasticity estimations are interpreted at the mean carbon price of the intervention during the period studied by the primary study.

Standard errors are derived accordingly. If the statistical (in)significance of an estimate at a specified significance level is the only uncertainty measure provided, this information is used to approximate the standard error. For the non-linear effect size transformation in the case of log-level regression coefficients, we derive the standard error by keeping the t statistic constant. For effect sizes from price elasticity estimations, we interpret the standard errors at the mean price level, just as for the transformation of the effect size itself.

### Effect size averaging
We use a multilevel random effects model to estimate the average treatment effect. The random effects model does not assume that all effect sizes converge to a common effect size mean[82], which in our case accounts for the heterogeneity in the studied countries and schemes. The common variance component is estimated using the restricted maximum likelihood (REML) estimation[83,84]. We apply a multilevel estimation to account for the non-independence of effect sizes from the same study, assuming a compound symmetric variance-covariance matrix[84].

For the estimation of average treatment effects for the individual policy schemes we extend the random effects model to a mixed effects model, inserting dummy variables for each carbon pricing scheme. Studies conducting a cross-sectional assessment of a set of carbon pricing schemes in multiple countries are collected with a separate dummy variable. The eight Chinese pilot ETS schemes are collected in a single dummy variable, as they are commonly assessed together as a single policy in the primary studies. For many of the schemes only one to five studies are available, which does not allow for appropriate clustering of the effect sizes[85,86]. The multilevel estimation of the model should still adequately capture the non-independence of effect sizes from the same study. Clustering of standard errors would have a marginal impact on the standard errors derived for the full sample averages (see Supplementary Information). The models are estimated in R using the metafor package[84].

To check that no single study exerts undue influence on the average effect sizes measured, we calculate Cook's distance and DFBETAS. For three studies in the sample the values of these metrics are distinctly different. All three studies assess the effect of emissions from the burning of coal. As these effects likely result from fuel switching without capturing the overall emission effect, 13 effect sizes from five studies with a focus on emissions from coal are excluded in the main assessment. Estimates including these studies are provided in the Supplementary Information, resulting in an average treatment effect of −12.5%.

To correct for publication bias, we follow the guidance by Stanley et al.[87] and Ioannidis et al.[38] and estimate the model for a reduced set of the adequately powered effect sizes. To assess the power of each effect size we use the standard error of each effect and assume the genuine effect to be the average treatment effect from our full set random effects model. We follow common practise and assume studies with power of above 80% to be adequately powered[88]. We estimate a multilevel random effects model, in line with our main approach, instead of a fixed effects model proposed in the literature[38,87].

## Heterogeneity assessment

There is considerable heterogeneity in the effect sizes ($I^2$=0.86 in the random effects model). To capture the variation in the response to the policy, we code variables for the carbon pricing schemes as well as information on the sector coverage of the scheme (or the study, where the study focuses on a single sector), the mean carbon price level during the assessment period, and a variable distinguishing carbon taxes from cap-and-trade schemes. The information on sector coverage and the price level was added from external sources[53,89]. We furthermore code a set of variables on the study design, estimation methods, and data used from the primary studies. Details on the moderator variables are provided in the Supplementary Information.

Given the large number of potential explanatory variables, we use the Bayesian model averaging technique (BMA)[22,90–92], employing a Markov chain Monte Carlo (specifically, the Metropolis-Hastings algorithm of the bms package for R[93]) to walk through the most likely combinations of explanatory variables. In the baseline specification we employ the unit information prior which is recommended by Eicher et al.[94]. This agnostic prior reflects our lack of knowledge regarding the probability of individual parameter values. To test the robustness of our estimates we follow Havranek et al.[22,92] and use the dilution prior that adjusts model probabilities by multiplying them by the determinant of the correlation matrix of the variables included in the model. Furthermore, as another robustness check, we follow Ley and Steel and apply the beta-binomial random model prior, which gives the same weight to each model size[95], as well as Fernández at al. who use the so-called BRIC g-prior[96]. The BMA results using alternative priors are provided in the Supplementary Information.

## Reporting summary

Further information on research design is available in the Nature Portfolio Reporting Summary linked to this article.

# Data availability

The study and effect size data collected for this study have been deposited in Github and can be accessed here: https://github.com/doebbeling/carbon_pricing_effectiveness.git.

# Code availability

The code used for the meta-analysis has been deposited in Github and can be accessed here: https://github.com/doebbeling/carbon_pricing_effectiveness.git.

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

## Acknowledgements

N.D.H. is supported by a PhD stipend from the Heinrich Böll Stiftung. J.C.M, T.M.K., O.E., N.K., M.C., C.F., and M.K. acknowledge research funding from the German Ministry of Education and Research (ARIADNE project–Grant No. 03SFK5J0). J.C.M., K.M., and M.K. acknowledge funding from the European Union under the Horizon program (CAPABLE project—Grant No. 101056891). Views and opinions expressed are however those of the authors only.

## Author contributions

J.C.M., N.D.H., K.M., T.M.K., O.E., N.K., W.F.L., N.O. and J.C.S. designed the research. N.D.H., K.M., M.C. and J.C.M. developed the literature screening strategy. N.D.H., K.M., M.B., N.K., W.F.L., N.O., J.C.S. and J.C.M. manually screened the literature and N.D.H., K.M. and M.B. extracted the data. N.D.H. and M.C. performed the machine learning-enabled screening. N.D.H., K.M., T.M.K. and S.B.B. performed the meta-analysis. T.M.K. and N.D.H. conducted the Bayesian Model Averaging analysis. N.D.H., K.M., T.M.K., S.B.B., O.E., C.F., P.M.F., M.K., N.K., J.C.S. and J.C.M. analysed the results. N.D.H., T.M.K., J.C.M. and K.M. wrote the manuscript with contributions from all authors.

## Competing interests

The authors declare no competing interests.
