## [Peer Review File · Nature Communications]

Systematic review and meta-analysis of ex-post evaluations on the effectiveness of carbon pricingReviewers' Comments:

Reviewer #1:

Remarks to the Author:

Referee of paper " Effectiveness of carbon pricing – A systematic review and meta-analysis of the ex-post literature"

Summary.

This is a very important area of work and it is positive and interesting to see a thorough piece of new work in the shape of a meta analysis that analyzes many earlier articles. The results are intuitively appealing and interesting. There is a lot of potential for a great publication here. I am however quite concerned about three-four methodological issues.

I do not see the point of looking at average effects when the effect must depend on the size of the price (in addition to other variables such as other policies, time horizon, sectors included etc).

To date one of the most important experiences of actual taxation that has made a difference is the taxation of gasoline (and diesel) which across the world is probably more important than the schemes that are officially called carbon taxes and ETS (even though the latter are growing over time). This experience has been much analyzed earlier and there are detailed accounts of the price elasticities etc. I think this should be referenced and preferably integrated into the work,

I found the most interesting part was section 5 that starts to explain the results. I think that could be heavily expanded at the cost of earlier sections (which deal with an average that is in my mind quite fictitious and meaningless since we don't know the price). The current section 5 is however poorly explained and needs a lot more elucidation.

A fourth point is that the study focuses on an object called "a Carbon policy" – could be a tax or an ETS but we don't know what other policies are being implemented – there could be additional fees and regulations against fossil use – or the opposite hidden or not hidden subsidies. I believe it is the total price of fuels containing carbon compared to their alternatives that matters...

Below are the points that occurred to me while reading the manuscript – so they are a mixture of minor and bigger points. The tone is critical because – well that is partly given by the process – but I want to emphasize again that I found the work promising and interesting and my critique can hopefully inspire you to improve the work.

Detailed comments

1. Almost all European (and some other) countries have gigantic carbon taxes on gasoline (and sometimes diesel). I think this partial tax on the transport system is worth more attention and think the authors should pay more attention to this – since it is a very large "experiment" or case and they do not deal with it. It has earlier been studied extensively – there are literally many hundreds of studies of the price response and these studies should be acknowledged, referenced and discussed since they are exactly on the core of the subject discussed although with another name.

2. Sweden's carbon tax is listed as being only buildings and transport. This is not true – it also affects industry although at a lower and more complicated set of rates.

3. The methodology described lines 61-66 with thousands of studies in a database is not very convincing to me – we have no idea of what is in there, how these studies were coded and analysed. This exposes the analyst to considerable bias. If there were strange things or errors in all this data – then surely there is the risk that the results will be biased or even nonsense. There is no effort made to describe the data or allay these fears. I get the impression the authors are proud to have a large number of studies but for me this is a disadvantage. If these data are not described and I don't know anything about their quality or purpose then – why should I trust the numbers. In this contest the large number of studies is not necessarily an advantage. In the last section of section 2, I read that many of these appear unpublished, they do not even have elasticities and half of them are for China where the schemes are called "carbon price" but in reality the price is close to zero and there are so many strange institutional details about these programs that in fact it is not clear if they all deserve to be counted. As an example I believe most of the Chinese schemes are intensity based rather than emissions based.

4. On line 73-75 we have something that looks like a main results "We find consistent evidence that carbon pricing policies have caused emission reductions. With the introduction of the policy, a sustained reduction of 10.4% is achieved on average (6.8% when correcting for publication bias)". I find this result a little absurd. As if "carbon pricing" were one type of an event like a solar eclipse or something – but surely the result will depend on the SIZE of the carbon tax (and other factors). .. I don't see any point in working out an average effect (with or without publication bias (the latter is a nice detail). I would stick to always reporting results that relate the effect to the size of the tax – as for instance elasticities.

5. With such heterogeneity I am not sure I would report a global average at all – I certainly would not do it with four significant figures "10.42" and all the trappings of a scientific result like error bands etc. It seems to me that is a treatment one should reserve for programs that are well defined and somewhat more homogeneous.

6. The idea of dealing with quality bias and publication bias is a good one and I appreciate this aspect. The trouble is however that not having the carbon price is a bigger problem. Could it not be the case that the studies of more dubious quality are the same ones where you did not have a carbon price?

7. Another problem is that the comparison – and the underlying studies – deal with highly disparate and heterogeneous objects. Why should in fact the effect sizes or their elasticities be similar in transport and in different kinds of industry or buildings?

8. Much of the paper, I think up to and including section 4 is about average effect sizes – which as I mention above, seems to me a totally strange concept. We are comparing the effects of a tax of say 100 dollars in some sectors with one of a few dollars in some other sector. How are we supposed to even start to interpret this?

9. Section 5 finally does start to analyze differences and does include interesting variables such as the size of the carbon price, its duration and the sectors covered. This is in principle the part that would be most interesting. However I found it short and ill explained full of acronyms and complicated methods that are not well defined. I am really interested and work in the broad area but could not make much out of figure 2 nor table 2.

10. The authors say they cannot assess price elasticities formally because there are too few of them and they recognize the limitations of their work on page 11 in a way that I find realistic and promising. They also speak of mixed methods and other data. I would like to suggest one body of literature and experience that is not used and seems to me relevant: There is a gigantic literature on the elasticities of demand for transport fuel – mainly gasoline, but also others like diesel. It is possible that earlier authors have over-emphasised this sector but it has several advantages from an analytic viewpoint. Gasoline is almost homogeneous and road based transport vehicles also have similarities. The difference between prices in different countries is largely due to policy – ie taxes. Sometimes these are called carbon taxes and sometimes they are called something else – but what counts is the price. Some countries in your sample say they have a carbon tax or ETS but at the same time they have several other policies including subsidies that are not being analysed. When you study the elasticity of gasoline demand you capture the total effect of all policies irrespective of what they are called which should be what we are really interested in. This literature as I mentioned in bullet point one. Of course I acknowledge that this literature is partial and only covers one sector – it is however the sector where there has been been the most (price-based) policy and It seems to me that it would be worth mentioning. The actual carbon reduction is much bigger than most other national carbon schemes mentioned, see for instance Sterner 2007. If there are indeed sophisticated mixed-method approaches it could perhaps even be included in the analysis.

References

Sterner, T. (2007). Fuel Taxes: An important instrument for climate policy. *Energy Policy*, 35(6), 3194-3202. DOI: <https://doi.org/10.1016/j.enpol.2006.10.025>

Reviewer #2:

Remarks to the Author:

Dear Authors,

I have read the abovementioned manuscript and find it very interesting and needed in the literature since there has been a lot of debate recently on this topic without sufficiently good evidence. I think your piece may fill this important gap, and therefore I congratulate you with this endeavour.

However, while reading your paper I encountered a few important drawbacks which must be addressed first before the paper can be published. Please see my details comments below:

Major comments

1. Literature overview. I feel you overlooked a few important discussion studies in your literature overview.
 - a. You mention under number 13 the paper by Martinez Alvarez et al arguing that carbon price is difficult to sustain over long term, but forget to mention van den Bergh and Savin 2023a who argued that empirical base for such a conclusion was incorrect
 - b. Similarly, when you mention the review by Lilliestam et al. under number 23 you omit its critique by van den Bergh and Savin 2021 showing how incomplete their review was
 - c. Point b has been also extended in a more recent study by the same authors in van den Bergh and Savi 2023b. I think those studies are worth mentioning to show different opinions among economists on effectiveness of carbon pricing
 - d. Another study next to Rosenbloom (number 9) stressing lack of infrastructure (for low carbon energy and transport) as barrier for carbon pricing is by Savin et al. (2020)
2. At few places in your manuscript you promise to provide materials but do not do it yet. I think the materials should be provided and checked before acceptance, not after. Therefore, please provide working links for your code, data, full query and review protocol. These will be kept confidential.
3. I know you mentioned that measurement of carbon price elasticity of emission has been undertaken by few studies only. Still I expected you to devote a section or a couple of paragraphs to summarising those finding with the caveat that those results may not be representative. You instead decided to omit this completely. I find this unfortunate and suggest to reconsider the decision.
4. While in Section 5 Table 2 you tried to distinguish a bit between effects of carbon tax and carbon market on emissions, I feel this is insufficient. First, because Table 2 and Figure 2 are relatively hard to read. Second, because compared to all other distinctions, this is by far the most popular and critical for policy analysis ; what is better a flexible carbon market or a stable carbon tax. This discussion has been started long ago by Goulder and Schein (2013) and more recently extended by Foramitti et al. (2021) using numerical models, but to date a good empirical comparison was missing. With your dataset you have a chance to give a more clear answer to that, but you seem to avoid stating this strongly. I wonder why. And if you could elaborate more on this

Minor:

1. In abstract and later in the paper be specific about the period at which 70 carbon pricing periods have been implemented. We live in a rapidly changing world, and in a year this number can change considerably. The paper revision instead may sometimes take long
2. Table 1: if British Columbia covers all sectors, why emission coverage is 78%? Perhaps good to explain to the reader
3. You say that more than 40 carbon pricing schemes have not yet been evaluated. Can you provide the list of them similar to your Table 1?
4. Specify in Figure 3 what are "other" exclusion criteria that removed 14 more studies

References:

1. J. van den Bergh, I. Savin, Political leadership, climate policy, and renewable energy. Proc Natl.

Acad. Sci. U.S.A. (2023a).

2. van den Bergh, Jeroen and Savin, Ivan, Impact of Carbon Pricing on Deep Decarbonisation: A Rejoinder to Lilliestam Et Al. (2022) (2023b). <http://dx.doi.org/10.2139/ssrn.4352574>

3. van den Bergh, J., Savin, I. Impact of Carbon Pricing on Low-Carbon Innovation and Deep Decarbonisation: Controversies and Path Forward. *Environ Resource Econ* 80, 705–715 (2021). <https://doi.org/10.1007/s10640-021-00594-6>

4. Savin, I., Drews, S., Maestre-Andrés, S. et al. Public views on carbon taxation and its fairness: a computational-linguistics analysis. *Climatic Change* 162, 2107–2138 (2020). <https://doi.org/10.1007/s10584-020-02842-y>

5. Goulder, L.H.; Schein, A. Carbon Taxes Versus Cap and Trade: A Critical Review. *Clim. Chang. Econ.* 2013, 4, 1–28

6. Foramitti, J.; Savin, I.; van den Bergh, J.C.J.M. Emission tax vs. permit trading under bounded rationality and dynamic markets. *Energy Policy* 2021, 148, 112009.

Reviewer #3:

Remarks to the Author:

Referee Report

“Effectiveness of carbon pricing – A systematic review and meta-analysis of the ex-post literature”

Overview

The paper conducts a meta-analysis of ex-post carbon pricing evaluations to quantify the impacts of carbon pricing policies on emission reductions. Based on 483 effect sizes extracted from 80 causal evaluations across 21 carbon pricing schemes, the paper finds that: 1) introducing a carbon price causes emission reductions of about 10.4% on average (6.8% after correcting for publication bias). Compared to similar studies that summarize the findings of the primary carbon pricing research, this paper contributes: 1) a quantitative synthesis of primary research findings; 2) a formal methodology for harmonizing primary research; 3) a critical appraisal of the quality of the primary research considered.

Methodology

A) The paper transforms the effect sizes extracted from different primary studies into a common effect metric: the percentage difference between the observed emissions and counterfactual emissions without carbon pricing. To estimate the average treatment effect, the paper uses a multilevel random effects model (which does not assume that all effect sizes converge to a common mean and thus accounts for the heterogeneity in the primary studies). The common variance component is estimated by the restricted maximum likelihood (REML) estimation.

B) Identification of potential bias in primary research such as unreasonable selection of a control group; inadequate control for confounding factors; from statistical specifications that do not allow to single out the policy effect. Although 46% of the studies are assessed to have a medium or high risk of bias, the average treatment effect is practically unchanged when removing those that are regarded as risky.

C) Correction of publication bias (a tendency towards publishing statistically significant effects): correct for publication bias by estimating average effects for a subsample of effect sizes with adequate statistical power.

D) Explaining heterogeneity in effect sizes: The paper uses Bayesian model averaging (BMA) to assess such heterogeneity. Explanatory variables include: dummy variables for each of the carbon pricing schemes, price level, sector coverage, and a variable capturing whether the carbon pricing policy was a carbon tax or ETS, type of study design, estimation method, and data used in the primary studies, etc (see the BMA result below). The posterior inclusion probability (PIP) indicates the relevance of each variable, providing the share of regression models in which the variable was included.

Based on the BMA result, the empirical analysis reached the following conclusions:

I. There are statistically significant differences in effect sizes across schemes in various geographies. For instance, The variables for the RGGI and the Chinese ETS pilots have a larger negative treatment effect than the EU ETS (line 207-215).

COMMENT: Naturally, different carbon pricing policies in different areas have different emission reduction effects. What is the point of including the dummy variables of each carbon pricing scheme? Maybe it is interesting to explore why different primary research on the same ETS/carbon tax got different emission reduction results, but why does the paper need to explain the heterogeneity across countries? In addition, how does the inclusion of dummy variables of each carbon pricing scheme help the author achieve this goal? This part is very confusing and the author needs to further explain the rationale behind the BMA as the chosen explanatory variables.

II. The paper includes variable "duration" to test whether primary research assessing longer periods after the policy finds higher treatment effects. Based on the coefficient before "duration," the paper finds that studies assessing the effectiveness over longer time periods find larger emission reductions (line 216-219).

COMMENT: This conclusion is not flawless: if a carbon pricing policy leads to larger emission reductions over a longer time, then the heterogeneity is not caused by methodological choices of the primary research. To single out the effects of methodological choices, the author needs to additionally prove that the emission reduction effects of a carbon pricing policy are constant/decreasing over time.

Data

The paper searches the bibliographic databases Web of Science, Scopus, JSTOR, RePEc, and Google Scholar for quantitative ex-post study designs. It studies 80 ex-post evaluations across 21 carbon pricing schemes around the world: 46 studies on the pilot emission trading schemes in China, 13 studies on the EU ETS, 7 on British Columbia, and 5 on the Regional Greenhouse Gas Initiative (RGGI) in the US.

Comments and recommendations

The paper addresses an important and timely topic. It provides new evidence in an area in which robust evidence is badly needed to inform the action of policy-makers.

However, in my opinion, the work as it currently stands raises a number of questions that the authors should address to render the proposed evidence sufficiently robust and to convince the reader that the paper worths being published.

A) What is the significance of conducting a meta-analysis for effects of carbon pricing across different regions? As countries have different socioeconomic backgrounds, the emission reduction effects tend to vary a lot. It is probably more reasonable to conduct a meta-analysis for one single region instead of averaging differences among heterogeneous regions. The author needs to further explain why conducting a meta-analysis is appropriate in this context. In addition, when averaging the emission reduction effects, the paper does not distinguish between an ETS and a carbon tax. As these two carbon pricing policies are based on different rationales and adopted by different countries, it is more appropriate to assess their effects separately and analyze whether there exists synergy/counteraction between them.

B) The paper mentions several times that the estimated emission reduction effects (10.4% reduction with variations across schemes ranging from 5% to 21%) are substantially larger than the 0-2% suggested in the recent and widely cited review by Green (2021). If the paper intends to argue against the 0-2% result, it needs to formally analyze the drawbacks of Green's study in effect harmonization method and selection of primary study and show why its own approach is more scientific and balanced. Among the primary studies selected by the paper, more than half of them are about China's pilot ETS pilots. This is not a very balanced sample and the author can consider selecting same amount of ex-post evaluations for each ETS/carbon tax based on research quality (eg. low risk of bias, similar methodology choice, etc.).

C) If possible, the author should include more mathematical details about "Standardising effect sizes" and "Effect size averaging" in the main body of the paper rather than in the Supplementary

Information.

REVIEWER COMMENTS

Reviewer #1 (Remarks to the Author):

Referee of paper " Effectiveness of carbon pricing – A systematic review and meta-analysis of the ex-post literature"

Summary.

This is a very important area of work and it is positive and interesting to see a thorough piece of new work in the shape of a meta analysis that analyzes many earlier articles. The results are intuitively appealing and interesting. There is a lot of potential for a great publication here. I am however quite concerned about three-four methodological issues.

Thank you for the positive feedback and all the valuable points raised.

I do not see the point of looking at average effects when the effect must depend on the size of the price (in addition to other variables such as other policies, time horizon, sectors included etc).

Thank you for raising this concern. We aim to strike a balance in the review between the assessment of the policy effectiveness across the schemes and the effectiveness of single schemes. As outlined in the introduction of the manuscript, there exists a debate in the academic literature whether carbon pricing is effective in general. These discussions are frequently picked up in science assessments such as those by the IPCC. We therefore believe that it is important to look at both and highlight the effectiveness across schemes, while emphasising at the same time that (considerable) heterogeneity across different schemes is expected as well as observed. But we agree that the presentation of the average effect across schemes certainly needs to be treated with caution. We therefore went carefully through the text and amended the wording around the presentation of average effect to highlight that it is just an average across the diverse set of policies. Moreover, we added a justification for the perspective across schemes to section 2.1, reordered the presentation of the results in sections 2.2 and 2.3 to provide more emphasis on the country specific results, and further caveat this point in the discussion.

Section 2.1

Beyond these differences in policy design, carbon price levels, and regional contexts, all considered policy experiences speak to the question whether carbon pricing is or is not effective in reducing GHG emissions. A systematic assessment and comparison of the outcomes of these policies can inform policymakers and future research by synthesising the available evidence.

Section 2.2

As depicted in Panel a of Fig. 1, emissions reduction effects are observed consistently across schemes with considerable variation in magnitude. We find average reduction effects from the introduction of a carbon price from about -21% to about -5%, with most of these being statistically significant. Across carbon pricing schemes, we find that on average the policy has reduced emissions by -10.4% [95% CI = (-11.9%, -8.9%)]. This effect is both substantial and highly statistically significant.

Section 2.3

When we remove studies with medium or high risk of bias from the sample, the average treatment effects for some of

the schemes are adjusted by up to 5 percentage points, while the estimation uncertainty increases due to the reduction of considered primary estimates (see Fig. 1, Panel d). The identified biases, however, do not systematically impact the estimated treatment effects in either direction.

Section 2.3

This subsample analysis adjusts most of the schemewise average treatment effects towards lower estimated emission reductions (see Fig. 1, Panel g). Across the schemes, the average treatment effect is reduced to -6.8% [95% CI = (-8.1%, -5.6%)]. Despite these adjustments, the publication bias corrected estimates support the overall finding that carbon pricing policies cause significant reductions in in GHG emissions.

Discussion

The synthesis of research findings across carbon pricing schemes provides comprehensive and consistent evidence of its effectiveness, despite the heterogeneity of policy designs and regional contexts. The average treatment effect across policies provides a numeric answer to the question, how effective carbon pricing policies have been, on the basis of the entire evidence available to us. This summary statistic should, however, not be generalised to other unstudied or planned carbon pricing policies. We highlight the presence of substantial variation across the schemes in our sample, ranging from 5% for the carbon tax in British Columbia to 21% for the RGGI (excluding estimates derived from studies with a high risk of bias). Transferring this finding to new policy proposals needs to consider the specific contextual factors.

To date one of the most important experiences of actual taxation that has made a difference is the taxation of gasoline (and diesel) which across the world is probably more important than the schemes that are officially called carbon taxes and ETS (even though the latter are growing over time). This experience has been much analyzed earlier and there are detailed accounts of the price elasticities etc. I think this should be referenced and preferably integrated into the work,

Thank you for pointing out this omission in our manuscript. While we had to draw the system boundaries somewhere, we had not appropriately acknowledged this relevant literature in our article. We are grateful for your very relevant comment and added reference to this literature in our introduction and discussion. To maintain a certain level of comparability between the reviewed policies, we were unable to include other pricing instruments into our assessments, even though they would provide further valuable insights.

Introduction

One way to assess the effects of carbon pricing is to evaluate experiences in the real world. A growing scientific literature has provided quantitative evaluations of the effects of different carbon pricing schemes on emissions [14, 15, 16], employment [17, 18], investments in low carbon technologies [19, 20] or on different societal groups [21, 22] amongst others mainly using causal statistical methods. Beyond the assessments of explicit carbon pricing policies, there are large bodies of literature studying the environmental outcomes of taxes on specific fuels [23, 24, 25, 26], of fossil fuel subsidies [27, 28, 29] or of the so called effective carbon rate, a combined price measure of carbon prices and fuel taxes [30, 31, 32, 33, 34, 35]. These ex-post evaluations provide an opportunity to ground the debate on carbon pricing in evidence.

Discussion

Further primary research is needed to estimate carbon price elasticities. In the meantime this debate could also be advanced by systematically exploring price elasticities estimated with respect to other pricing instruments. There has been a long history of evaluating taxes on fossil fuel consumption [23, 24, 25, 26]. Efforts have been made to combine these different instruments into a combined measure of the price rate per emitted ton of carbon [30, 31, 32, 33, 34, 35]. While this meta-analysis has focussed on explicit carbon taxes and cap-and-trade systems, we acknowledge the large potential to learn on market-based instruments from this wider literature.

I found the most interesting part was section 5 that starts to explain the results. I think that could be heavily expanded at the cost of earlier sections (which deal with an average that is in my mind quite fictitious and meaningless since we don't know the price). The current section 5 is however poorly explained and needs a lot more elucidation.

Thank you for this suggestion. We have expanded our explanations in that section. In particular, we have added an extended introduction to the section, outlining the importance of heterogeneity assessments for our data and referencing to three relevant debates on factors which are expected to influence the effectiveness of carbon pricing policies: the sectors covered by the scheme, the prices vs. quantities debate and the size of the carbon price.

There is considerable variation in the effect sizes reported by primary studies included in this review. This could arise from heterogeneity in the design of the carbon pricing policies or from heterogeneity in the design of the primary studies. The carbon pricing literature mainly discusses three policy design factors that could potentially explain differences in the effectiveness of the policy. First, there are debates whether carbon prices are better applied as carbon taxes or as emission trading schemes [54, 55, 56, 57, 58]. Secondly, it is argued that the policy causes different reduction rates in different sectors [59, 60, 61]. And thirdly, the level of the carbon price can be expected to play a decisive role for the magnitude of the emission reductions [5, 62, 63]. We assess whether, and to what extent, such factors are able to explain differences in the treatment effects reported. We test which factors are most relevant to explain the reported emission reductions by using scheme and study characteristics as explanatory variables in meta-regressions.

We furthermore provide the reader a better guidance in how the outcomes of our models can be interpreted.

The results from the BMA are provided in Table 2 and Fig. 2. The posterior inclusion probability (PIP) indicates the relevance of each variable. Commonly, variables with a PIP above 0.5 are interpreted to be relevant explanatory factors, while variables with lower PIPs are unable to capture the observed heterogeneity. The table furthermore provides the estimated coefficient and standard deviation averaged across all estimated meta-regressions.

This is also supported by enhanced captions for Table 2 and Figure 2, such that the presented results are easier to grasp for the readers.

Table 2: The table provides the results of meta-regressions using Bayesian model averaging. The posterior inclusion probability (PIP) indicates the relevance of each variable. Variables with $PIP \geq 0.5$ are considered relevant for explaining the heterogeneity in carbon emissions reductions reported across primary studies. Post Mean and Post SD

represent the mean and standard deviation of the posterior distribution for a respective explanatory variable. Five variables have $PIP \geq 0.5$ and are considered relevant: the dummy variables for RGGI, Chinese pilot ETS, Swiss ETS, Data City, and duration. The dummy variables represent the geographic location in which the policy was implemented, with the reference location being EU ETS. Data City captures whether primary studies used city level data versus country level data. The variable duration captures the number of years for which data on the scheme was collected after the policy was implemented. Definitions of the other explanatory variables are provided in the Supplementary Information.

Figure 2: Heterogeneity assessment using Bayesian model averaging: The columns in the figure depict the best 26,435 estimated meta-regressions, with each column showing the outcome of one estimated meta-regression model. The dependent variable for each of the meta-regression models is the percentage change in emissions. The possible explanatory variables are depicted in the rows (ordered by their PIP in descending order) and the explanatory variables included in a respective meta-regression model of the column is indicated by the colours. Red colour indicates the variable was included with a negative sign (larger emission reductions). Blue colour indicates a positive sign (smaller emission reductions). No colour indicates that the variable was not included in the meta-regression model represented by that column. The horizontal axis indicates the cumulative posterior model probabilities across all models. The models are ranked by their posterior model probability with the model on the left accounting for the largest posterior model probability. The definitions of the explanatory variables are provided in the Supplementary Information.

Providing the reader with a better understanding of the method, in turn, allows us to be more clear in the presentation of the findings. The section benefits from these improvements, as we communicate now more clearly, what conclusions can be drawn from the assessment and where the method is unable to provide a clear answer. These are also picked up more extensively in the discussion now.

A fourth point is that the study focuses on an object called “a Carbon policy” – could be a tax or an ETS but we don’t know what other policies are being implemented – there could be additional fees and regulations against fossil use – or the opposite hidden or not hidden subsidies. I believe it is the total price of fuels containing carbon compared to their alternatives that matters...

You are right that there is a multitude of (market based) policy factors that plays a role for the emission levels. We here concentrate on explicit carbon pricing policies in the form of taxes or cap-and-trade schemes as these are increasingly considered and implemented by policymakers (in addition to the other policies you name). This introduction of a new policy measure is empirically interesting to explore and politically interesting to understand as an additional policy option. In the reviewed primary studies we critically appraised whether the authors reasonably control for such other policy factors. This certainly does not mean that policies are directly transferable to other countries with a different policy context. We agree that there is a lot of additional literature on market based climate policies that should also be synthesised, which was unfortunately beyond our capabilities for this meta-analysis.

We caveat this more clearly now in our discussion, pointing to the challenges involved in isolating the effect of single policies.

The multitude of supplementary or conflicting policies as well as other confounding factors pose a challenge to the clear identification of the causal effects of a specific policy [84]. Novel methods of reverse causality are a promising avenue to address this challenge [85].

We furthermore reference to other empirical literature assessing other pricing mechanisms on fossil fuel use as an outlook for further synthesis research required.

Below are the points that occurred to me while reading the manuscript – so they are a mixture of minor and bigger points. The tone is critical because – well that is partly given by the process – but I want to emphasize again that I found the work promising and interesting and my critique can hopefully inspire you to improve the work.

Thank you for all your valuable comments.

Detailed comments

1. Almost all European (and some other) countries have gigantic carbon taxes on gasoline (and sometimes diesel). I think this partial tax on the transport system is worth more attention and think the authors should pay more attention to this – since it is a very large “experiment” or case and they do not deal with it. It has earlier been studied extensively – there are literally many hundreds of studies of the price response and these studies should be acknowledged, referenced and discussed since they are exactly on the core of the subject discussed although with another name.

In response to this comment and the comments above we have included references to the suggested literature. Thank you for pointing to this fundamental gap in our manuscript. As outlined above, we unfortunately had to draw the system boundaries for our review somewhere to be able to make the study results quantitatively comparable. We discuss this caveat in our discussion section now and reference to the relevant literature, which we agree should be synthesised next to complete the picture.

2. Sweden's carbon tax is listed as being only buildings and transport. This is not true – it also affects industry although at a lower and more complicated set of rates.

You are right. We should have been more explicit that we simplified the information for the presentation in the table. We make this more explicit in the caption now.

All information on the carbon pricing schemes was retrieved from the World Bank [3]. The information for the sector coverage was simplified. For more detailed information on the coverage, including covered or exempted subsectors, the reader is referred to the World Bank data.

3. The methodology described lines 61-66 with thousands of studies in a database is not very convincing to me – we have no idea of what is in there, how these studies were coded and analysed. This exposes the analyst to considerable bias. If there were strange things or errors in all this data – then surely there is the risk that the results will be biased or even nonsense. There is no effort made to describe the data or allay these fears. I get the impression the authors are proud to have a large number of studies but for me this is a disadvantage. If these data are not described and I don't know anything about their quality or purpose then – why should I trust the numbers. In this context the large number of studies is not necessarily an advantage. In the last section of section 2, I read that many of these appear unpublished, they do not even have elasticities and half of them are for China where the schemes are called “carbon price” but in reality the price is close to zero and there are so many strange institutional details about these programs that in fact it is not clear if they all deserve to be counted. As an example I believe most of the Chinese schemes are intensity based rather than emissions based.

Thank you for expressing your doubts about this methodological approach. It highlights in particular that our communication of the same needs to be improved. We have amended the mentioned

section to make more explicit that from the 16,000 documents we found 80 to be relevant for our review.

We fill this gap by conducting a systematic review and meta-analysis of the empirical ex-post literature on the effectiveness of carbon pricing, covering 21 enacted carbon pricing policies around the globe following the guidelines by the Collaboration for Environmental Evidence [45]. We use a machine-learning enhanced approach as proposed by Callaghan and Müller-Hansen [46] to screen 16,748 studies from five different literature databases, identifying 80 relevant ex-post policy assessments. We extract and harmonise estimates of average emissions reductions from the introduction of a carbon price. We conduct a meta-analysis on 483 effect sizes on 21 different carbon pricing schemes and estimate emission reduction effects.

The reason for mentioning the number is to emphasise the efforts taken to search for relevant documents, as a comprehensive search is a key requirement for systematic reviews provided by the various guidelines (including the guidelines by the Collaboration for Environmental Evidence, we follow here). Our comprehensive search and machine-learning assisted screening of the literature have proven very effective for this review, as we have identified more than twice as many relevant primary studies as considered in any previous review of the literature on carbon pricing effectiveness. In the methods section and particular in Figure 3 we provide further information on the screening of these documents. In addition we have now added the research protocol to the supplementary information, which contains further information on the search and inclusion of studies.

We agree with the reviewer that assessing and controlling for study quality is critical. This is often neglected in reviews and meta-analyses. Here we perform a high-standard publication bias assessment – as frequently required in the economics literature. But we also assess the risk of bias for each study included and also control for quality moderators in our heterogeneity analysis (but there are no systematic effects on the results). Our analysis therefore carefully deals with these aspects and, in fact, highlights a considerable share of studies with a high risk of bias. For the specific case of the Chinese pilot ETS schemes, these are covered by our review as they are considered carbon pricing schemes by the World Bank.

We have also rephrased the last paragraph of section 2 (now 2.1) as we perceive that this was not clear and caused some confusion. Most of the 80 papers are published and peer reviewed. Only a hand full are working papers.

Our systematic review also reveals some fundamental evidence gaps in the literature. Despite the broad set of bibliographic databases searched, we found evidence only for 20 out of 73 carbon pricing policies in place in 2023 [3] and for the Australian carbon tax, which was repealed two years after its implementation. For some, more recently implemented, policies this may be explained by the time needed for sufficient data to become available, be assessed, and the results published. But even of the 38 carbon pricing schemes already implemented by 2015, for 18 of these we could not find a single study on effectiveness, despite the broad set of bibliographic databases searched. There is also little evidence on the effectiveness of carbon pricing relative to the level of the carbon price (carbon price elasticity). We identify only nine price elasticity studies, providing too few effect sizes for meta-analysing these separately.

4. On line 73-75 we have something that looks like a main results “We find consistent evidence that carbon pricing policies have caused emission reductions. With the introduction of the policy, a

sustained reduction of 10.4% is achieved on average (6.8% when correcting for publication bias)”. I find this result a little absurd. As if “carbon pricing” were one type of an event like a solar eclipse or something – but surely the result will depend on the SIZE of the carbon tax (and other factors). .. I don’t see any point in working out an average effect (with or without publication bias (the latter is a nice detail). I would stick to always reporting results that relate the effect to the size of the tax – as for instance elasticities.

Thank you for sharing your concerns regarding the presentation of this as a main finding. We have revised the sentence to make explicit that this reduction summarises the experience of the 21 reviewed policies.

With the introduction of the policy, a sustained reduction of 10.4% is achieved on average across the reviewed carbon pricing schemes (6.8% when correcting for publication bias).

We have furthermore revised the results sections in order to present the averages more carefully, increasing the weight of the average effects for each of the schemes and considered whether we should emphasise the schemewise results already in the introduction more explicitly. We however came to the view that a simplified representation of our results is valuable in the introduction and the more complex assessment follows in the subsequent sections. We therefore left it with the general statement following that sentence:

Our heterogeneity analysis suggests substantial variation in the average effect across the schemes implemented in various geographies.

Throughout the manuscript we have revisited the language on the presentation of the average effect across the schemes emphasising that it is just the average across the evaluated policies. Elasticities can unfortunately not be constructed from the available evidence. We added a more explicit discussion of this in the discussion section.

While we would have been keen to analyse the effects of changing carbon prices on emissions reductions, we found an insufficient number of studies to formally assess the carbon price elasticity. We identified only nine primary studies estimating such elasticities. Four are using the stepwise introduction of the carbon tax in British Columbia to estimate elasticities for the transport and buildings sectors [14, 64, 68, 69], while one is conducted respectively for RGGI [70] and EU ETS [71]. In addition, some studies estimate elasticities across countries and carbon pricing schemes [72, 73, 15]. Instead of synthesising this limited set of studies separately, we were able to harmonise the effect sizes of eight of these studies to treatment effects and assess them together with the other studies in our sample.

While we controlled for carbon price levels as part of our heterogeneity analysis, we do not identify a relationship between the price level and the achieved emission reductions. This is not very surprising given our comparative research design focused on treatment effects and a comparison of different carbon pricing schemes around the globe, which are applied in heterogeneous regional contexts with diverse abatement costs [74].

Further primary research is needed to estimate carbon price elasticities. In the meantime this debate could also be advanced by systematically exploring price elasticities estimated with respect to other pricing instruments. There has been a long history of evaluating taxes on fossil fuel consumption [23, 24, 25, 26]. Efforts have been made to combine these different instruments into a combined measure of the price rate per emitted ton of carbon [30, 31, 32, 33, 34, 35].

While this meta-analysis has focussed on explicit carbon taxes and cap-and-trade systems, we acknowledge the large potential to learn on market-based instruments from this wider literature.

5. With such heterogeneity I am not sure I would report a global average at all – I certainly would not do it with four significant figures “10.42” and all the trappings of a scientific result like error bands etc. It seems to me that is a treatment one should reserve for programs that are well defined and somewhat more homogeneous.

We agree with the reviewer that we should do all efforts to present the average effect with caution. We have amended the wording around the presentation of the average to account for that. We also take the advice not to report the finding with 4 significant digits. We have reduced all findings to one decimal place. However we find it relevant to present the average including its error bands. The averaging is based on an estimation method. Presenting the outcome without uncertainties would also not be good scientific practise. We would like to emphasise that we do not interpret this result as a universally applicable estimate of the effectiveness of the policy, but as the average effect of the reviewed studies and policies.

6. The idea of dealing with quality bias and publication bias is a good one and I appreciate this aspect. The trouble is however that not having the carbon price is a bigger problem. Could it not be the case that the studies of more dubious quality are the same ones where you did not have a carbon price?

Thank you for sharing your concerns about the quality of the primary studies. We understand that you are referring to quality differences between price elasticity studies and those studies that estimate a treatment effect. Fortunately, our rigorous quality appraisal of the studies revealed that there is a considerable amount of studies which performed sound policy assessments. Most studies use some form of difference-in-differences designs or synthetic control methods. These methods are well established for policy impact evaluations. After critically appraising all studies we are convinced that this study design can equally be applied to carbon pricing policies. Of course, we notice that these designs do not make full use of all information (i.e. the price level), but are still able to capture the effect of the policy as a treatment effect that reduces emissions at the time of the introduction of the policy and is constant in the following years. These research designs depend on the use of good treatment and control groups and controlling for confounding factors. It rests on the assumption that the policy is a stable treatment, which particularly implies in our case that carbon prices are unchanged. Since most of the studies assess time frames of up to 5 years after the policy was introduced, this assumption, to our understanding, is a defensible simplification. In our set we find 54% of the studies apply sound research designs.

The prevalence of such research designs however does not provide further insights into the dependence of the reductions on the price levels. This largely remains a research gap.

7. Another problem is that the comparison – and the underlying studies – deal with highly disparate and heterogenous objects. Why should in fact the effect sizes or their elasticities be similar in transport and in different kinds of industry or buildings?

As stated above, we try to balance between synthesising the evidence and highlighting also the differences between the studies and policies. In response to your comments we have tried to put more emphasis on these arguments throughout the text. However, we would argue that the research question, whether and to what extent carbon pricing policies are effective, which is still contested in the literature, needs to be answered across the heterogeneous applications of the policy. This is also what IPCC assessments have been focussing on. We therefore feel that our manuscript links to this more general literature and guides readers then towards the important carbon pricing scheme -

specific results. As highlighted before, we have edited the manuscript throughout to reflect your concerns and avoid confusion.

8. Much of the paper, I think up to and including section 4 is about average effect sizes – which as I mention above, seems to me a totally strange concept. We are comparing the effects of a tax of say 100 dollars in some sectors with one of a few dollars in some other sector. How are we supposed to even start to interpret this?

We have restructured the text to put more emphasis on the findings for each of the pricing schemes, which as you argue are the more relevant findings as compared to an overall average.

9. Section 5 finally does start to analyze differences and does include interesting variables such as the size of the carbon price, its duration and the sectors covered. This is in principle the part that would be most interesting. However I found it short and ill explained full of acronyms and complicated methods that are not well defined. I am really interested and work in the broad area but could not make much out of figure 2 nor table 2.

Thank you for flagging this shortcoming in our manuscript. We have substantially improved the section in response to your comment. Most importantly, we have improved our explanations of the method to guide the reader through the text. We have improved the captions to table 2 and figure 2. With these improvements on the technical side we are in a better position to explain our findings of the heterogeneity section. We provide an enhanced introduction to the section outlining the variables we are most interested in. We pick up the discussion regarding the most interesting variables in the discussion section again.

10. The authors say they cannot assess price elasticities formally because there are too few of them and they recognize the limitations of their work on page 11 in a way that I find realistic and promising. They also speak of mixed methods and other data. I would like to suggest one body of literature and experience that is not used and seems to me relevant: There is a gigantic literature on the elasticities of demand for transport fuel – mainly gasoline, but also others like diesel. It is possible that earlier authors have over-emphasised this sector but it has several advantages from an analytic viewpoint. Gasoline is almost homogeneous and road based transport vehicles also have similarities. The difference between prices in different countries is largely due to policy – ie taxes. Sometimes these are called carbon taxes and sometimes they are called something else – but what counts is the price. Some countries in your sample say they have a carbon tax or ETS but at the same time they have several other policies including subsidies that are not being analysed. When you study the elasticity of gasoline demand you capture the total effect of all policies irrespective of what they are called which should be what we are really interested in. This literature as I mentioned is vast and there are many large surveys of this literature. I mentioned this already in bullet point one. Of course I acknowledge that this literature is partial and only covers one sector – it is however the sector where there has been the most (price-based) policy and It seems to me that it would be worth mentioning. The actual carbon reduction is much bigger than most other national carbon schemes mentioned, see for instance Sterner 2007. If there are indeed sophisticated mixed-method approaches it could perhaps even be included in the analysis.

Thank you for bringing this up. We have added a reference to other empirical literature on market-based climate policies in the discussion and reference to this literature now also in the introduction. However, our analysis already goes well beyond what has been reviewed before and it is out of scope to include it in our formal analysis. That said, we share your enthusiasm for this literature and earmark this for future work. In the discussion we caveat this point and highlight this as a relevant field requiring synthesis.

References

Sterner, T. (2007). Fuel Taxes: An important instrument for climate policy. *Energy Policy*, 35(6), 3194-3202. DOI: <https://doi.org/10.1016/j.enpol.2006.10.025>

Reviewer #2 (Remarks to the Author):

Dear Authors,

I have read the abovementioned manuscript and find it very interesting and needed in the literature since there has been a lot of debate recently on this topic without sufficiently good evidence. I think your piece may fill this important gap, and therefore I congratulate you with this endeavour.

Thank you for the positive feedback and all the valuable points raised.

However, while reading your paper I encountered a few important drawbacks which must be addressed first before the paper can be published. Please see my details comments below:

Major comments

1. Literature overview. I feel you overlooked a few important discussion studies in your literature overview.

a. You mention under number 13 the paper by Martinez Alvarez et al arguing that carbon price is difficult to sustain over long term, but forget to mention van den Bergh and Savin 2023a who argued that empirical base for such a conclusion was incorrect

Thank you for pointing us to these relevant discussions around that article. While considering the publication more closely it became particularly apparent that the cited study is not dealing with explicit carbon pricing but with gasoline taxes. As these are not the subject of this review, we decided to remove the reference to Martinez Alvarez et al.

b. Similarly, when you mention the review by Lilliestam et al. under number 23 you omit its critique by van den Bergh and Savin 2021 showing how incomplete their review was

Thank you for pointing us to this relevant omission. We have added a reference to the suggested study. This important debate has also partly motivated us to do this systematic review, though focussing on emission reductions instead of innovation.

c. Point b has been also extended in a more recent study by the same authors in van den Bergh and Savi 2023b. I think those studies are worth mentioning to show different opinions among economists on effectiveness of carbon pricing

Thank you for suggesting this additional reference. In the introduction and motivation of the article we try to be rather concise. We therefore do not see the room to dive deeper into this very relevant debate.

d. Another study next to Rosenbloom (number 9) stressing lack of infrastructure (for low carbon energy and transport) as barrier for carbon pricing is by Savin et al. (2020)

Thank you for pointing us to this additional reference. We have added it to the suggested part of our manuscript.

2. At few places in your manuscript you promise to provide materials but do not do it yet. I think the materials should be provided and checked before acceptance, not after. Therefore, please provide working links for your code, data, full query and review protocol. These will be kept confidential.

Thank you for requesting the additional material. The data and code are provided in the supplementary material to this submission. For the resubmission we have added the protocol. Please excuse the missing links. These were removed to comply with the double-anonymous review guidance.

3. I know you mentioned that measurement of carbon price elasticity of emission has been undertaken by few studies only. Still I expected you to devote a section or a couple of paragraphs to summarising those finding with the caveat that those results may not be representative. You instead decided to omit this completely. I find this unfortunate and suggest to reconsider the decision.

Thank you for the suggestion. We have extended the discussion of this in the discussion section as follows:

While we would have been keen to analyse the effects of changing carbon prices on emissions reductions, we found an insufficient number of studies to formally assess the carbon price elasticity. We identified only nine primary studies estimating such elasticities. Four are using the stepwise introduction of the carbon tax in British Columbia to estimate elasticities for the transport and buildings sectors [14, 64, 68, 69], while one is conducted respectively for RGGI [70] and EU ETS [71]. In addition, some studies estimate elasticities across countries and carbon pricing schemes [72, 73, 15]. Instead of synthesising this limited set of studies separately, we were able to harmonise the effect sizes of eight of these studies to treatment effects and assess them together with the other studies in our sample.

While we controlled for carbon price levels as part of our heterogeneity analysis, we do not identify a relationship between the price level and the achieved emission reductions. This is not very surprising given our comparative research design focused on treatment effects and a comparison of different carbon pricing schemes around the globe, which are applied in heterogeneous regional contexts with diverse abatement costs [74].

Further primary research is needed to estimate carbon price elasticities. In the meantime this debate could also be advanced by systematically exploring price elasticities estimated with respect to other pricing instruments. There has been a long history of evaluating taxes on fossil fuel consumption [23, 24, 25, 26]. Efforts have been made to combine these different instruments into a combined measure of the price rate per emitted ton of carbon [30, 31, 32, 33, 34, 35].

While this meta-analysis has focussed on explicit carbon taxes and cap-and-trade systems, we acknowledge the large potential to learn on market-based instruments from this wider literature.

4. While in Section 5 Table 2 you tried to distinguish a bit between effects of carbon tax and carbon market on emissions, I feel this is insufficient. First, because Table 2 and Figure 2 are relatively hard to read. Second, because compared to all other distinctions, this is by far the most popular and critical for policy analysis ; what is better a flexible carbon market or a stable carbon tax. This discussion has been started long ago by Goulder and Schein (2013) and more recently extended by Foramitti et al. (2021) using numerical models, but to date a good empirical comparison was missing. With your dataset you have a chance to give a more clear answer to that, but you seem to avoid stating this strongly. I wonder why. And if you could elaborate more on this

We were hoping to be able to provide a clearer answer to this question, but unfortunately our results do not provide an answer to that. We recognise that this was not stated clear enough in our article so far. We have worked on the heterogeneity section to make this more clear. We motivate our heterogeneity analysis more explicitly by linking to this previous literature.

There is considerable variation in the effect sizes reported by primary studies included in this review. This could arise from heterogeneity in the design of the carbon pricing policies or from heterogeneity in the design of the primary studies. The carbon pricing literature mainly discusses three policy design factors that could potentially explain differences in the effectiveness of the policy. First, there are debates whether carbon prices are better applied as carbon taxes or as emission trading schemes [54, 55, 56, 57, 58]. Secondly, it is argued that the policy causes different reduction rates in different sectors [59, 60, 61]. And thirdly, the level of the carbon price can be expected to play a decisive role for the magnitude of the emission reductions [5, 62, 63]. We assess whether, and to what extent, such factors are able to explain differences in the treatment effects reported. We test which factors are most relevant to explain the reported emission reductions by using scheme and study characteristics as explanatory variables in meta-regressions.

We pick this up in the discussion again.

The magnitude of the achieved emission reductions differs substantially across the different carbon pricing schemes. The reviewed evidence, however, does not provide an answer regarding which national factors best explain the variation in the observed effect sizes. The evidence on 21 carbon pricing policies is unable to resolve the debate whether taxes or caps on emission quantities should be favoured or provide new evidence on differences in the emission reductions achieved in different sectors of the economy. We acknowledge the limitations of our purely quantitative, meta-econometric approach here – which is inherently dependent on the available evidence base.

We also amended the captions to the mentioned figure and table to be easier to understand. Thank you for pointing us to this shortcoming.

Table 2: The table provides the results of meta-regressions using Bayesian model averaging. The posterior inclusion probability (PIP) indicates the relevance of each variable. Variables with $PIP \geq 0.5$ are considered relevant for explaining the heterogeneity in carbon emissions reductions reported across primary studies. Post Mean and Post SD represent the mean and standard deviation of the posterior distribution for a respective explanatory variable. Five variables have $PIP \geq 0.5$ and are considered relevant: the dummy variables for RGGI, Chinese pilot ETS, Swiss ETS, Data City, and duration. The dummy variables represent the geographic location in which the policy was implemented, with the reference location being EU ETS. Data City captures whether primary studies used city level data versus country level data. The variable duration captures the number of years for which data on the scheme was collected after the policy was implemented. Definitions of the other explanatory variables are provided in the Supplementary Information.

Figure 2: Heterogeneity assessment using Bayesian model averaging: The columns in the figure depict the best 26,435 estimated meta-regressions, with each column showing the outcome of one estimated meta-regression model. The dependent variable for each of the meta-regression models is the percentage change in emissions. The possible explanatory variables are depicted in the rows (ordered by their PIP in descending order) and the explanatory variables included in a respective meta-regression model of the column is indicated by the colours. Red colour indicates the variable was included with a negative sign (larger emission

reductions). Blue colour indicates a positive sign (smaller emission reductions). No colour indicates that the variable was not included in the meta-regression model represented by that column. The horizontal axis indicates the cumulative posterior model probabilities across all models. The models are ranked by their posterior model probability with the model on the left accounting for the largest posterior model probability. The definitions of the explanatory variables are provided in the Supplementary Information.

Minor:

1. In abstract and later in the paper be specific about the period at which 70 carbon pricing periods have been implemented. We live in a rapidly changing world, and in a year this number can change considerably. The paper revision instead may sometimes take long

We have updated the numbers with the latest update of the World Bank's update of the Carbon Pricing Dashboard and added the year reference to the text in the second section of the paper, where we discuss the policies more deeply. For the benefit of the flow of the text we prefer not to add the time reference to the abstract and the introduction. In the introduction the reference to the source is also provided, which indicates the reference year.

Our systematic review also reveals some fundamental evidence gaps in the literature. Despite the broad set of bibliographic databases searched, we found evidence only for 20 out of 73 carbon pricing policies in place in 2023 [3] and for the Australian carbon tax, which was repealed two years after its implementation.

2. Table 1: if British Columbia covers all sectors, why emission coverage is 78%? Perhaps good to explain to the reader

Thank you for spotting this mistake. We corrected the sectors in the list to "industry, power, transport and buildings". Emissions from AFOLU sectors are not included. This should explain the difference.

3. You say that more than 40 carbon pricing schemes have not yet been evaluated. Can you provide the list of them similar to your Table 1?

Thank you for your suggestion. We have added such a list to the Supplementary Information and reference to it in the respective sections of the main text.

4. Specify in Figure 3 what are "other" exclusion criteria that removed 14 more studies

We have removed the "other" category from the figure. Most of the studies could be allocated to other exclusion categories, while for some we now specified that they were not available in English.

References:

1. J. van den Bergh, I. Savin, Political leadership, climate policy, and renewable energy. *Proc Natl. Acad. Sci. U.S.A.* (2023a).
2. van den Bergh, Jeroen and Savin, Ivan, Impact of Carbon Pricing on Deep Decarbonisation: A Rejoinder to Lilliestam Et Al. (2022) (2023b). <http://dx.doi.org/10.2139/ssrn.4352574>
3. van den Bergh, J., Savin, I. Impact of Carbon Pricing on Low-Carbon Innovation and Deep Decarbonisation: Controversies and Path Forward. *Environ Resource Econ* 80, 705–715 (2021). <https://doi.org/10.1007/s10640-021-00594-6>
4. Savin, I., Drews, S., Maestre-Andrés, S. et al. Public views on carbon taxation and its fairness: a computational-linguistics analysis. *Climatic Change* 162, 2107–2138 (2020). <https://doi.org/10.1007/s10584-020-02842-y>

5. Goulder, L.H.; Schein, A. Carbon Taxes Versus Cap and Trade: A Critical Review. *Clim. Chang. Econ.* 2013, 4, 1–28

6. Foramitti, J.; Savin, I.; van den Bergh, J.C.J.M. Emission tax vs. permit trading under bounded rationality and dynamic markets. *Energy Policy* 2021, 148, 112009.

Reviewer #3 (Remarks to the Author):

Referee Report

“Effectiveness of carbon pricing – A systematic review and meta-analysis of the ex-post literature”

Overview

The paper conducts a meta-analysis of ex-post carbon pricing evaluations to quantify the impacts of carbon pricing policies on emission reductions. Based on 483 effect sizes extracted from 80 causal evaluations across 21 carbon pricing schemes, the paper finds that: 1) introducing a carbon price causes emission reductions of about 10.4% on average (6.8% after correcting for publication bias). Compared to similar studies that summarize the findings of the primary carbon pricing research, this paper contributes: 1) a quantitative synthesis of primary research findings; 2) a formal methodology for harmonizing primary research; 3) a critical appraisal of the quality of the primary research considered.

Methodology

A) The paper transforms the effect sizes extracted from different primary studies into a common effect metric: the percentage difference between the observed emissions and counterfactual emissions without carbon pricing. To estimate the average treatment effect, the paper uses a multilevel random effects model (which does not assume that all effect sizes converge to a common mean and thus accounts for the heterogeneity in the primary studies). The common variance component is estimated by the restricted maximum likelihood (REML) estimation.

B) Identification of potential bias in primary research such as unreasonable selection of a control group; inadequate control for confounding factors; from statistical specifications that do not allow to single out the policy effect. Although 46% of the studies are assessed to have a medium or high risk of bias, the average treatment effect is practically unchanged when removing those that are regarded as risky.

C) Correction of publication bias (a tendency towards publishing statistically significant effects): correct for publication bias by estimating average effects for a subsample of effect sizes with adequate statistical power.

D) Explaining heterogeneity in effect sizes: The paper uses Bayesian model averaging (BMA) to assess such heterogeneity. Explanatory variables include: dummy variables for each of the carbon pricing schemes, price level, sector coverage, and a variable capturing whether the carbon pricing policy was a carbon tax or ETS, type of study design, estimation method, and data used in the primary studies, etc (see the BMA result below). The posterior inclusion probability (PIP) indicates the relevance of each variable, providing the share of regression models in which the variable was included.

Based on the BMA result, the empirical analysis reached the following conclusions:

I. There are statistically significant differences in effect sizes across schemes in various geographies. For instance, The variables for the RGGI and the Chinese ETS pilots have a larger negative treatment effect than the EU ETS (line 207-215).

COMMENT: Naturally, different carbon pricing policies in different areas have different emission reduction effects. What is the point of including the dummy variables of each carbon pricing scheme? Maybe it is interesting to explore why different primary research on the same ETS/carbon

tax got different emission reduction results, but why does the paper need to explain the heterogeneity across countries? In addition, how does the inclusion of dummy variables of each carbon pricing scheme help the author achieve this goal? This part is very confusing and the author needs to further explain the rationale behind the BMA as the chosen explanatory variables.

Thank you for emphasising that the heterogeneity section needs to be strengthened. We have tried to improve that section to communicate aims and results more clearly. We specifically state more clearly now that we include the dummies for the different schemes to capture remaining contextual variables. If we did not include these variables we would risk that other explanatory variables capture these variations, comparable to omitted variable bias in primary research. Among other improvements to the section, it now includes this explanation:

We include explanatory variables for the three policy design factors provided above: price level, sector coverage, and a variable differentiating between carbon taxes and cap-and-trade schemes. In addition we add dummy variables for each of the carbon pricing schemes, capturing the remaining policy design and contextual factors of each policy scheme.

II. The paper includes variable “duration” to test whether primary research assessing longer periods after the policy finds higher treatment effects. Based on the coefficient before “duration,” the paper finds that studies assessing the effectiveness over longer time periods find larger emission reductions (line 216-219).

COMMENT: This conclusion is not flawless: if a carbon pricing policy leads to larger emission reductions over a longer time, then the heterogeneity is not caused by methodological choices of the primary research. To single out the effects of methodological choices, the author needs to additionally prove that the emission reduction effects of a carbon pricing policy are constant/decreasing over time.

Thank you for pointing out that our presentation of this finding allows for different interpretations. Our interest in the variable was indeed more towards the interpretation you are also pointing to in your comment. We cautiously interpret this finding as an indication for an increasing effectiveness of the policy with time. The data does not allow for a detailed assessments of (constant or decreasing) time trends in the estimated reductions. The largest part of the primary studies assumes a constant treatment effect for their estimations. Estimating time trends based on these primary estimates is not possible and would violate the assumptions they are derived on. We therefore do interpret this coefficient as an indication for increasing emission reductions with time, while being cautious due to the aforementioned reasons.

We have strengthened this argument in the discussion section.

For carbon taxes and emission trading schemes, the reviewed evidence suggests that the policy effectiveness increases with time. The meta-regressions reveal that studies conducted for longer time periods after the introduction of the carbon price report larger emission reduction effects compared to assessments for shorter time periods. While this finding should be treated with caution, as most of the primary studies assume constant treatment effects for their estimations, it hints towards increasing emissions reductions in the years following the policy introduction. The assumption of constant treatment effects reflects not only methodological considerations of the primary studies, but is also based on the expectation that as long as the carbon price of the implemented policy is unchanged, the emission reduction effect should not intensify. The finding of our meta-regression to some extent counters that assumption. An increasing policy effectiveness could be a result of steady adjustment processes, enforced by innovation and investments into cleaner production and infrastructure. Additionally, the literature reviewed here provides some evidence that an increasing policy stringency has also played its role in strengthening the effectiveness of

the policy. Increases in the carbon prices led to additional emission reductions in Sweden [75] and the United Kingdom [76]. Similar effects are found for the EU ETS, where the effectiveness increases with the increasing stringency from phases I, II, and III [77, 78, 79, 80].

Data

The paper searches the bibliographic databases Web of Science, Scopus, JSTOR, RePEc, and Google Scholar for quantitative ex-post study designs. It studies 80 ex-post evaluations across 21 carbon pricing schemes around the world: 46 studies on the pilot emission trading schemes in China, 13 studies on the EU ETS, 7 on British Columbia, and 5 on the Regional Greenhouse Gas Initiative (RGGI) in the US.

Comments and recommendations

The paper addresses an important and timely topic. It provides new evidence in an area in which robust evidence is badly needed to inform the action of policy-makers.

However, in my opinion, the work as it currently stands raises a number of questions that the authors should address to render the proposed evidence sufficiently robust and to convince the reader that the paper worths being published.

Thank you for your positive feedback and all the valuable comments.

A) What is the significance of conducting a meta-analysis for effects of carbon pricing across different regions? As countries have different socioeconomic backgrounds, the emission reduction effects tend to vary a lot. It is probably more reasonable to conduct a meta-analysis for one single region instead of averaging differences among heterogeneous regions. The author needs to further explain why conducting a meta-analysis is appropriate in this context. In addition, when averaging the emission reduction effects, the paper does not distinguish between an ETS and a carbon tax. As these two carbon pricing policies are based on different rationales and adopted by different countries, it is more appropriate to assess their effects separately and analyze whether there exists synergy/counteraction between them.

Thank you for raising this concern. We aim to strike a balance in the review between the assessment of the policy effectiveness across the schemes and the effectiveness of single schemes. As outlined in the introduction of the manuscript, there exists a debate in the academic literature whether carbon pricing is effective in general. These discussions are frequently picked up in science assessments such as those by the IPCC. We therefore believe that it is important to look at both and highlight the effectiveness across schemes, while emphasising at the same time that (considerable) heterogeneity across different schemes is expected as well as observed. In response to your comment, we added a justification for the perspective across schemes to section 2.1, reordered the presentation of the results in sections 2.2 and 2.3 to provide more emphasis on the country specific results, and further caveat this point in the discussion.

Section 2.1

Beyond these differences in policy design, carbon price levels, and regional contexts, all considered policy experiences speak to the question whether carbon pricing is or is not effective in reducing GHG emissions. A systematic assessment and comparison of the outcomes of these policies can inform policymakers and future research by synthesising the available evidence.

Section 2.2

As depicted in Panel a of Fig. 1, emissions reduction effects are observed consistently across schemes with considerable variation in magnitude. We find average reduction effects from the introduction of a carbon price from about

-21% to about -5%, with most of these being statistically significant. Across carbon pricing schemes, we find that on average the policy has reduced emissions by -10.4% [95% CI = (-11.9%, -8.9%)]. This effect is both substantial and highly statistically significant.

Section 2.3

When we remove studies with medium or high risk of bias from the sample, the average treatment effects for some of the schemes are adjusted by up to 5 percentage points, while the estimation uncertainty increases due to the reduction of considered primary estimates (see Fig. 1, Panel d). The identified biases, however, do not systematically impact the estimated treatment effects in either direction.

Section 2.3

This subsample analysis adjusts most of the schemewise average treatment effects towards lower estimated emission reductions (see Fig. 1, Panel g). Across the schemes, the average treatment effect is reduced to -6.8% [95% CI = (-8.1%, -5.6%)]. Despite these adjustments, the publication bias corrected estimates support the overall finding that carbon pricing policies cause significant reductions in in GHG emissions.

Discussion

The synthesis of research findings across carbon pricing schemes provides comprehensive and consistent evidence of its effectiveness, despite the heterogeneity of policy designs and regional contexts. The average treatment effect across policies provides a numeric answer to the question, how effective carbon pricing policies have been, on the basis of the entire evidence available to us. This summary statistic should, however, not be generalised to other unstudied or planned carbon pricing policies. We highlight the presence of substantial variation across the schemes in our sample, ranging from 5% for the carbon tax in British Columbia to 21% for the RGGI (excluding estimates derived from studies with a high risk of bias). Transferring this finding to new policy proposals needs to consider the specific contextual factors.

B) The paper mentions several times that the estimated emission reduction effects (10.4% reduction with variations across schemes ranging from 5% to 21%) are substantially larger than the 0-2% suggested in the recent and widely cited review by Green (2021). If the paper intends to argue against the 0-2% result, it needs to formally analyze the drawbacks of Green's study in effect harmonization method and selection of primary study and show why its own approach is more scientific and balanced. Among the primary studies selected by the paper, more than half of them are about China's pilot ETS pilots. This is not a very balanced sample and the author can consider selecting same amount of ex-post evaluations for each ETS/carbon tax based on research quality (eg. low risk of bias, similar methodology choice, etc.).

Thank you for highlighting that the reader would be interested to better understand the difference between our findings and Green's findings. The unfortunate answer to this is that the review by Green is not providing any methodology for the provided numbers provides. It simply presents the 0-2% as a summary of studies. We are therefore unable to formally compare our findings. We have considered to make this critique more central in our manuscript, but have decided against that, as we believe that our analysis provides many new insights that could be lost in a methodological debate. We however consider to follow up on these methodological questions in a separate paper, where we have more space to outline the methodological differences. In this paper we have

amended the reference in the discussion section to indicate that we are unable to formally compare our findings to Green's.

These policies have caused emissions reductions of 10.4% on average with substantial variations across schemes ranging from 5% to 21%. These are substantially larger than the 0-2% suggested in the recent and widely cited review by Green [39], which lacks a clear and transparent methodology to synthesise the literature [40], not allowing us to formally compare our results.

Thank you also for your second suggestion on a more balanced sample of studies for each of the schemes. In line with the common guidance for systematic reviews and meta-analyses (we here follow the guidelines by the Collaboration for Environmental Evidence) all relevant primary studies should be considered. We have considered how we could best implement your suggestion as a robustness test. While we appreciate the general idea of creating a more balanced sample, we are concerned of creating other biases if we aim for drastically reducing the number of studies on the Chinese ETS pilots, while leaving low-quality studies for other schemes in the sample. Our risk of bias assessment found 32 of the Chinese studies to be of adequate quality, while, for instance, for the Swiss ETS the only study in our sample had a high risk of bias. Instead we test the robustness of our findings when excluding all Chinese studies and find that the average reduction effect is slightly reduced to -8.14%. We include this robustness check in our supplementary information.

C) If possible, the author should include more mathematical details about “Standardising effect sizes” and “Effect size averaging” in the main body of the paper rather than in the Supplementary Information.

We have considered your suggestion and opted not to overload the methods section with mathematical formulas. While we see the value of being even more explicit about the methods, we want to keep the section concise and accessible. We make all code used for the standardisation and meta-analysis available with the study. We hope with this we strike a good balance between accessibility and conciseness on the one hand and transparency and replicability on the other hand.

Reviewers' Comments:

Reviewer #1:

Remarks to the Author:

The Manuscript has improved - but only marginally.

Most of my comments revolve around the fact that the effect of a carbon tax depends on its size. The authors reply very politely ofcourse and say thank you and make some (good) editorial changes but have not made very fundamental changes.

They have not used the largest body of evidence from fuel taxes.

They still put quite a lot of emphasis on average taxes and though they have started to make some concessions to common sense, their paper is still not fundamentally restructured to focus on the size of carbon taxes. Hence we are still taking averages of programs where the carbon tax (or ETS price) is insignificantly low with programs where it is very high.

As an author I would have dropped these parts completely but at least I think they need to be downsized and put in context much more than now. The main results must somehow take into effect the size of carbon taxes. My preferred method is through elasticities but i suppose one could also find other measures, looking at different price increase categories and average effects of these if this is easier.

I still think there is potential for a valuable contribution but it requires a more significant reworking of the analysis.

Reviewer #2:

Remarks to the Author:

I thank the authors for detailed revision.

Obviously it is difficult to incorporate so many different comments and suggestions from three different referees, but I htink they did a rather good job.

I do not hve any further pending comments

Reviewer #3:

Remarks to the Author:

I went carefully through the revised version of the paper and the detailed accompanying letter prepared by the authors.

I praise their effort in addressing the numerous points raised by all reviewers and in explaining their replies in the letter.

I think the paper has substantially improved during the revision process.

REVIEWER COMMENTS

Reviewer #1 (Remarks to the Author):

The Manuscript has improved - but only marginally.

Most of my comments revolve around the fact that the effect of a carbon tax depends on its size.

Thank you for being so persistent in your request for us to appreciate the differences between the schemes more explicitly. To accommodate your comments, we have removed the reference to the average effect from the abstract, the introduction and the discussion sections. We instead summarise our findings now as follows:

Abstract:

Based on 483 effect sizes extracted from 80 causal ex-post evaluations across 21 carbon pricing schemes, we find that introducing a carbon price has yielded immediate and substantial emission reductions for at least 17 of these policies, despite the low level of prices in most instances. Statistically significant emissions reductions range between 5% to 21% across the schemes (4% to 15% after correcting for publication bias).

Introduction:

We find consistent evidence that carbon pricing policies have caused emissions reductions. Statistically significant emissions reductions are found for 17 of the reviewed carbon pricing policies, with immediate and sustained reductions of between 5% to 21% (4% to 15% when correcting for publication bias). Our heterogeneity analysis suggests that differences in estimates from the studies are driven by the policy design and context in which carbon pricing is implemented, while often discussed factors like cross-country differences in carbon prices, sectoral coverage, and the design of the policy as a tax or trading scheme do not capture the identified heterogeneity in effect sizes.

Discussion:

In this first quantitative meta-analysis of carbon pricing evaluations, we find robust evidence that existing carbon pricing schemes that have been evaluated to date are effective in reducing GHG emissions. Our machine-learning enhanced approach to study identification finds more than twice as many ex-post evaluations than existing reviews [15, 36, 37, 39], studying the effectiveness of 21 carbon pricing policies. Our meta-analysis finds that at least 17 of these policies have caused significant emissions reductions ranging from 5% to 21%. These are substantially larger than the 0-2% suggested in the recent and widely cited review by Green [39], which lacks a clear and transparent methodology to synthesise the literature [40], not allowing us to formally compare our results. Our finding is robust to biases from poor study designs as well as publication bias. Correcting for the latter adjusts the range of observed emissions reductions to 4% to 15% across carbon pricing schemes.

The synthesis of research findings across carbon pricing schemes provides comprehensive and consistent evidence of its effectiveness, despite the heterogeneity of policy designs and regional contexts. Compared to the recent assessment report by the IPCC, which provides a quantification of achieved reductions only for the EU ETS [64], our systematic review adds synthesised emission reduction estimates for more than a dozen carbon pricing schemes. We provide these estimates together with uncertainty ranges and a transparent assessment of study quality and highlight the presence of substantial variation in emission reductions

achieved across the schemes in our sample, ranging from 5% for the carbon tax in British Columbia to 21% for the RGGI. We conduct an early application of Bayesian model averaging for meta-regressions on our dataset of 483 effect sizes to disentangle which factors explain these differences. The findings suggest that the individual context and policy design of the schemes best explain the heterogeneity in achieved emissions reductions. These are the most relevant explanatory factors despite controlling for broader policy design features like the sectoral coverage or the design as carbon tax or carbon trading scheme as well as for study design features of the primary studies.

More importantly, we assess the role of prices more explicitly now. We conduct a more in depth assessment of the relationship between the price and the reduction effect using our BMA framework. You will notice that the assessment of carbon price levels across countries is not in line with economics intuition that higher prices would cause larger emissions reductions. For the interpretation of this finding it is important to stress that in our data only 9% of the total variation in the price variable is within scheme variation, while 91% is between scheme variation. We are therefore mainly able to study the between scheme price variation. In our discussion we provide a number of explanations, why across schemes and countries the assumption of a relationship between the price and the achieved emissions reductions is less clear and likely distorted by the multi-layered differences in the country contexts. In our supplementary information we add an additional BMA meta-regression assessment, where we remove the dummies controlling for the schemes. In that case the price variable even comes out with a positive sign, indicating that lower emissions reductions are achieved with higher prices. This counterintuitive finding is addressed in the discussion section, where we discuss example schemes from our study where larger (smaller) emissions reductions are achieved with lower (higher) prices.

We also would like to emphasise that we have screened the literature thoroughly for studies looking at carbon price elasticities using our machine-learning enhanced approach to literature search and selection. We find an insufficient number of studies to systematically analyse the effect of carbon price levels on emissions reductions and highlight this in the discussion as an important gap in primary ex-post evaluation studies on carbon taxes and emission trading schemes.

The discussions in the results section 2.4 and in the discussion section now read as follows:

Section 2.4:

Variation in carbon prices, the sectoral coverage of schemes, and choice of carbon tax vs. cap-and-trade do not seem to be important variables in explaining the observed heterogeneity in emissions reductions ($PIP < 0.5$). Instead the dummy variables for the place where the schemes are applied do a better job in explaining this heterogeneity than the variables that capture specific design characteristics. The variables for the RGGI and the Chinese ETS pilots have a larger reduction effect on emissions than the EU ETS, which is set as the reference category. The Swiss ETS is estimated to have less of a reduction effect compared to the benchmark. Alternative specifications of the BMA, provided in the Supplementary Information, also estimate a larger reduction effect for the Swedish carbon tax compared to the benchmark. The directions of these coefficients are in line with the average treatment effects presented in Fig. 1, for the respective geographies.

If we remove the dummy variables for the schemes, the size of the carbon price becomes an important variable in the BMA to explain the heterogeneity in emission reductions with a PIP close to 1 (see Supplementary Information). However, in the absence of the scheme dummies the effect of the price variable is likely to be confounded as the scheme dummies account for any omitted context variable that does not vary within a scheme. The high

correlation of 0.96 between the scheme dummies and the price variable indicates that the price variable captures the heterogeneity between schemes. In fact, the price coefficient is estimated with a positive sign in the BMA specification without the scheme dummies, implying that lower emission reductions are achieved with higher carbon prices. The counterintuitive direction of the price effect indicates a misspecification of the model when the scheme dummies are excluded. Below we discuss possible causes for this inverse relationship between the price and the reduction effect in our data. The effect of carbon prices on emission reductions is better identified by adding scheme dummies to focus on the variation of prices within each scheme. However, the largest share of the variation in our carbon price variable comes from variation between the schemes (91%) and only 9% from within scheme variation. This is not a limitation of our dataset. Indeed, carbon prices tend to vary strongly across countries based on the design and coverage of scheme. But for individual schemes prices have historically been stagnant (EU ETS till recently, RGGI, Chinese ETS pilots) or increases relatively modest (BC carbon tax) [48] and the effect size estimates evaluated here provide limited time frequency. We suspect that due to this low variation, our sample has insufficient power to identify carbon prices as a relevant factor in explaining emission reductions.

Discussion:

The synthesis of research findings across carbon pricing schemes provides comprehensive and consistent evidence of its effectiveness, despite the heterogeneity of policy designs and regional contexts. Compared to the recent assessment report by the IPCC, which provides a quantification of achieved reductions only for the EU ETS [65], our systematic review adds synthesised emission reduction estimates for more than a dozen carbon pricing schemes. We provide these estimates together with uncertainty ranges and a transparent assessment of study quality and highlight the presence of substantial variation in emission reductions achieved across the schemes in our sample, ranging from 5% for the carbon tax in British Columbia to 21% for the RGGI. We conduct an early application of Bayesian model averaging for meta-regressions on our dataset of 483 effect sizes to disentangle which factors explain these differences. The findings suggest that the individual context and policy design of the schemes best explain the heterogeneity in achieved emissions reductions. These are the most relevant explanatory factors despite controlling for broader policy design features like the sectoral coverage or the design as carbon tax or carbon trading scheme as well as for study design features of the primary studies.

Our heterogeneity analysis does not identify a relationship between the price level and the achieved emissions reductions. While economic intuition would suggest that higher carbon prices would lead to larger emissions reductions, with our set of 80 policy evaluations we do not find such a relationship. While a non-significant research finding does not provide evidence for the absence of such a relationship, the primary evidence reviewed here challenges whether a clear relationship between the carbon price level and the achieved emission reductions should be expected across carbon pricing policies. The reviewed literature finds larger emissions reduction effects for the pilot emission trading schemes in China (-13.1%) than for the EU ETS (-7.3%) and the carbon tax in British Columbia (-5.4%), despite the very low carbon prices of the Chinese schemes. The average prices of the eight Chinese pilot schemes are all below US\$ 8 during the study period, while the average prices for the EU and British Columbia are at US\$ 20 and US\$ 18, respectively. This is likely a result of lower abatement costs in China [66] together with differences in the policy contexts of the countries. The effectiveness is certainly influenced by other policies in place. In China indirect

carbon prices are lower than in the EU countries and Canada [67], allowing for a higher marginal effect of the implementation of the ETS pilots in China. Non-pricing instruments also diverge across countries. In addition, the implementation of a carbon price (even with a low price) can have a signalling effect towards the emitters, underlining the commitment of the government towards climate mitigation. Evidence for the Guangdong province suggests that signalling has significantly contributed to the achieved emissions reductions in the context of the introduction of the ETS pilots in China [68]. Another example highlighting the relevance of the context of the policy implementation is the case of the RGGI. The policy implementation coincides with the shale gas boom, which drastically reduced the prices of natural gas in the USA and started around the same time as the RGGI was implemented. In face of these general price dynamics in the US energy sector, RGGI participating states reduced their emissions considerably stronger compared to non-regulated states [69, 70], while the carbon price was only US\$ 3 on average.

Even if across schemes carbon price levels are not found as a statistically significant driver of the heterogeneity in the emissions reduction effects, within a scheme the effectiveness is expected to increase with increasing prices. While we would have been keen to analyse the effects of changing carbon prices on emissions reductions, we found an insufficient number of studies to formally assess the carbon price elasticity. We identified only nine primary studies estimating such elasticities. Four are using the stepwise introduction of the carbon tax in British Columbia to estimate elasticities for the transport and buildings sectors [14, 71, 72, 73], while one is conducted respectively for RGGI [70] and EU ETS [74]. In addition, some studies estimate elasticities across countries and carbon pricing schemes [75, 76, 15]. Instead of synthesising this limited set of studies separately, we were able to transform the elasticity estimates from eight of these studies to treatment effects and assess them together with the other studies in our sample. Estimating a price effect in our meta-analytic framework is constrained by the small within scheme variation of the price variable. 91% of the variation in carbon prices can be attributed to the different schemes while the variation within schemes only accounts for 9%, which is too small to reliably estimate the price effect.

The authors reply very politely ofcourse and say thank you and make some (good) editorial changes but have not made very fundamental changes.

They have not used the largest body of evidence from fuel taxes.

Thank you for bringing up the fuel tax literature once more. However, we would like to submit that this request is unreasonable as it overlooks the very substantial amount of work that went into this manuscript that goes far beyond any available study to date. The author team also does not agree substantively with the request, as we do not think that the two literature strands are directly comparable. Let me expand on both arguments below.

First, in terms of the practical infeasibility of the suggestion, we want to highlight that it has taken this large and experienced author team already almost two years to conduct this rigorous and transparent assessment of the carbon pricing literature. As such we want to emphasise the contribution of our work that goes far beyond what has been provided in the literature so far. Our review is the first review in the field that systematically screens the available empirical literature, resulting in an evaluation of more than twice as many studies as there were reviewed by any previous review. We are able to provide average emission reduction effects for 21 carbon pricing schemes, together with an estimated uncertainty and a transparent assessment of the quality of the primary evidence. The recent IPCC report provided such a quantification of the effectiveness only for

one carbon pricing scheme, the EU ETS. Using this comprehensive dataset of 483 emission reduction effects, we conduct a meta-regression analysis to explore the heterogeneity in the effect sizes. The use of Bayesian model averaging for meta-regression analysis is a recent advancement in the methods and is not yet regularly applied in the field. We are thus convinced that we not only provide relevant new insights for policy makers and further research, we also contribute to advancing the standards for evidence syntheses conducted in the field. As such the review carries – as is – a substantial amount of new synthetic evidence to the carbon pricing discourse that expands and backs the existing scientific knowledge on carbon pricing in multiple ways.

Second, after a series of discussions within the author team, we believe that the literature on (fossil) fuel taxes cannot be directly compared to the carbon pricing literature reviewed here. The largest part of the literature on gasoline taxes is concerned with the estimation of elasticities based on variations in the market prices of gasoline and not in tax induced variation in the prices. There is a growing literature, which shows that demand responses to volatile market-induced fuel price changes are much smaller than responses to fuel tax changes (Andersson, 2019; Davis and Kilian, 2011; Li et al., 2014; Rivers and Schaufele, 2015; Tiezzi and Verde, 2019; Zimmer and Koch, 2017). This may be explained by two reasons. First, tax increases are persistent price increases unlike market price volatility, which does not send a consistent price signal in a specific direction. Secondly, tax induced changes have a higher salience compared to market price changes, leading to larger changes in people's behaviour. Assessments of price elasticities of gasoline consumption thus describe a different effect than assessments of taxes or emission trading schemes. Systematically reviewing the gasoline price elasticity literature may certainly be relevant for other reasons, but analysing them together with carbon prices could lead to misleading conclusions.

They still put quite a lot of emphasis on average taxes and though they have started to make some concessions to common sense, their paper is still not fundamentally restructured to focus on the size of carbon taxes. Hence we are still taking averages of programs where the carbon tax (or ETS price) is insignificantly low with programs where it is very high.

As an author I would have dropped these parts completely but at least I think they need to be downsized and put in context much more than now. The main results must somehow take into effect the size of carbon taxes. My preferred method is through elasticities but i suppose one could also find other measures, looking at different price increase categories and average effects of these if this is easier.

As outlined above, we have changed the presentation of our findings to emphasise the number of carbon pricing schemes we find significant evidence for emissions reductions together with the range of emissions reductions observed for the schemes. The average effect across schemes is not emphasised anymore and only mentioned in the results section to summarise the data and to point out the agreement in the literature that if all evidence on carbon pricing schemes is taken together a statistically significant and environmentally considerable emission reduction is achieved.

To take into account the size of the carbon price, we add an alternative model specification to our BMA assessment. As this alternative specification shows large indication of model misspecification, we keep our previous model as the main specification and add the alternative specification as a robustness check in our Supplementary Information. We mention the counterintuitive finding in our results section and discuss the shortcomings of the model. In our discussion section we add an extensive discussion outlining why it is rather unsurprising that the price variable alone is unable to capture the differences between the schemes and that the variable instead captures the multi-

faceted differences between the schemes. These differences between the schemes are well described by our study. We present these differences in Figure 1, in our meta-regression analysis and our discussion section. To our knowledge, these have not been presented in such a comprehensive and comprehensible format before. Collecting the evidence on 21 carbon pricing schemes from 80 primary studies into a single source will be able to inform further research into which context factors may explain the differences and what role the size of the carbon price plays in the specific contexts.

Our findings on the inverse relationship between prices and reduction effects keep us from incorporating the size of the carbon price more directly into our quantitative synthesis. As there exists a correlation between the reduction effects and the size of the carbon price, with its sign opposite to economic theory, this relationship should not be overemphasised. This correlation is explained largely by other context variables and disappears in our BMA assessment when controlling for other context factors, in particular the dummy variables for the schemes (which can be interpreted like fixed effects in panel regressions). Therefore we oppose to use this single policy design characteristic to contextualise the effect sizes in the presentation of scheme-wise average effects. Presenting the effect sizes as a ratio of the price, for instance, would inflate those effects with high emission reductions and low prices further, while downsizing the effects of schemes with higher prices.

As stated in the manuscript, we would have also been keen to study carbon price elasticities, but find limited primary research studying price elasticities of carbon taxes and emission trading schemes. A transformation or interpretation of the treatment effects, we extracted from the large set of difference-in-differences studies, to elasticities is unfortunately not possible. It would require strong assumptions about the functional form of the relation (e.g. linear relationship) and thus not provide reliable elasticity estimates. In the main specification of our meta-regression we do not identify a relationship between the price and the reduction effect, likely resulting from the small within scheme variation of the prices (9% of the variation) compared to the between scheme variation (91%).

I still think there is potential for a valuable contribution but it requires a more significant reworking of the analysis.

Thank you for all the relevant comments. We are convinced that these are now reflected in our analysis.

References:

- Andersson, J.J., 2019. Carbon Taxes and CO₂ Emissions: Sweden as a Case Study. *Am. Econ. J. Econ. Policy* 11, 1–30. <https://doi.org/10.1257/pol.20170144>
- Davis, L.W., Kilian, L., 2011. Estimating the effect of a gasoline tax on carbon emissions. *J. Appl. Econom.* 26, 1187–1214. <https://doi.org/10.1002/jae.1156>
- Li, S., Linn, J., Muehlegger, E., 2014. Gasoline Taxes and Consumer Behavior. *Am. Econ. J. Econ. Policy* 6, 302–342. <https://doi.org/10.1257/pol.6.4.302>
- Rivers, N., Schaufele, B., 2015. Saliency of carbon taxes in the gasoline market. *J. Environ. Econ. Manag.* 74, 23–36. <https://doi.org/10.1016/j.jeem.2015.07.002>
- Tiezzi, S., Verde, S.F., 2019. The signaling effect of gasoline taxes and its distributional implications. *J. Econ. Inequal.* 17, 145–169. <https://doi.org/10.1007/s10888-018-9397-7>
- Zimmer, A., Koch, N., 2017. Fuel consumption dynamics in Europe: Tax reform implications for air pollution and carbon emissions. *Transp. Res. Part Policy Pract.* 106, 22–50. <https://doi.org/10.1016/j.tra.2017.08.006>

Reviewer #2 (Remarks to the Author):

I thank the authors for detailed revision.

Obviously it is difficult to incorporate so many different comments and suggestions from three different referees, but I think they did a rather good job.

I do not have any further pending comments

Reviewer #3 (Remarks to the Author):

I went carefully through the revised version of the paper and the detailed accompanying letter prepared by the authors.

I praise their effort in addressing the numerous points raised by all reviewers and in explaining their replies in the letter.

I think the paper has substantially improved during the revision process.

Reviewers' Comments:

Reviewer #1:

Remarks to the Author:

The authors have made some improvements in the paper. It is a pity that their method does not appear to be able to give any idea of the size of effects as in an elasticity. I note that you find results "ranging from 5% for the carbon tax in British Columbia to 21% for the RGG" These results are quite remarkable and strange. The price level in RGGI is just around say 6 USD while it is more like 65 (Canadian) dollars in BC: if you get a much bigger reaction from a much smaller price signal in RGGI you should ask yourself what is going on. Presumably there are a number of other confounding factors. Naturally producers and consumer react to the sum of all policies and market factors and a complete study should preferably try to account at least for the main ones.

As one example of the text we can take the following "The findings suggest that the individual context and policy design of the schemes best explain the heterogeneity in achieved emissions reductions. These are the most relevant explanatory factors despite controlling for broader policy design features like the sectoral coverage or the design as carbon tax or carbon trading scheme as well as for study design features of the primary studies." Well sure – context and design matter – but it seems to me that the size of the carbon price increase is the elephant in the room. Of course it is very likely that a big carbon price will have more effect than a small one....

However given the fact that your method is more focused on proving whether or not there is an effect at all, it is much better to remove numbers that are misleading. Numbers with "size effects" that are based on a mix of studies with widely varying price signals will just be confusing.

I still see however, at least one very substantial hurdle left to deal with before this paper is ready to make the kind of truly complete and important contribution that one would expect from a publication in Nature Communications.

I take the trouble to argue again about these issues because your subject is a very important one and your paper could be seminal and really important but to achieve this, careful consideration and careful writing is required. Let me first of all remind you that you do start your whole paper by saying:

"Countries are not on track to meet the climate goals they committed to under the Paris Agreement [1, 2]. To step up implementation, learning about what policy instruments work in reducing emissions at the necessary speed and scale is critical.

This is a tall order. It is one of humanity's big challenges and I think we agree that this issue is very important – academically and also for the sake of climate policy. As you know, I worry that you exclude a lot of literature that you don't see because you are just looking for objects that fit into the framework of an event study and carry the label "carbon price". I believe that firms and individuals react to the total price of an energy carrier like fuel oil or diesel (and other relevant factors like price of alternatives). If the price is high they use less and if the price is low they use more. The main difference across the world in prices is in fact due to policies like taxes. Sometimes these policies carry other names like gasoline tax, "import duty" or something else instead of carbon tax. However I don't think that the name of the tax is decisive for use – just its size – or rather the size of the resulting price on the market. I don't think motorists care what the minister said or if it was a minister rather than a monopoly that charged a high price.

When you speak of fuel elasticity studies you complain that they don't make a distinction if the price went up through a tax or through other market effects. I do agree there is a difference (and I think some studies have tried to address that although it is difficult) – but I am convinced this is a second order problem compared to not including any price at all or estimating together all kinds of "carbon tax events" whether they imply a big or a small price increase and thus getting no estimate of size. When it comes to really understanding "what policy instruments work in reducing emissions at the necessary speed and scale", I must say that I find studies that provide studies with an elasticity estimate much more useful than event studies that just give a yes or a no. I could agree however that there is room for both. Elasticity studies cannot always be readily performed and work better for sectors such as the transport fuels than for economy wide taxes. I can also partly sympathize with your argument that you have already done a lot of work, you are a large team of experienced

researchers, you have worked for more than 2 years already ... and basically that it would not be reasonable to ask you to do a whole lot of work of another kind. I am a little divided on this argument because of course work input does not guarantee publication in a good journal. You also have arguments about the size of elasticity estimates – “demand responses to volatile market-induced fuel price changes are much smaller than responses to fuel tax changes”. Well, maybe, However I am again not quite convinced – your own method does not give any size of elasticity estimates at all. Anyway I am intent on being constructive and do not want to fall in the category of a quarrelsome and one-sided referee. I think that we can agree that there are different methodological approaches. I think an overview of the kind of studies that you provide is useful. I am hoping you will upon reflection, agree that elasticity studies are useful too. Both have their advantages and their weaknesses. I tend to think that identifying the size of the price increase and the response it generates is very important and I think you find other factors to be more important. This divergence is ok and we can continue to reason about it. It is however imperative to give the reader a reasonable and fair description of the two methodologies and their respective advantages and disadvantages. At the moment you only have a short section starting with the words “Beyond the assessments of explicit carbon pricing policies...” which I find a little ideological and misrepresentative. There are many hundreds of elasticity studies and a very large number of surveys of these studies and they generally agree – within a reasonable margin – that price elasticities are quite high (in the long run). (it is not really my job to give a number but in the past I used to think that -0.7 was a reasonable value, that agreed with averages or meta-analyses of hundreds of studies).

In sum I regret the fact that you do now want to integrate this body of work into your own, I do think that a full scale integration and comparison would make a truly impactful and seminal piece, but I can see that this would be a lot of work and face some considerable challenges. I would therefore be prepared to accept that you don't do that for the present article. I do however think that this body of work – and this methodological approach is of such importance that it deserves a serious treatment, rather than a mere dismissal. In my opinion the paper needs, at the very least, to be complemented with such a serious description of this literature that says something of its advantages rather than just dismissing it as not scientific, not formal, very small effects etc that I do not find to be a serious or fair summary ... Considering that your method does not give effect sizes at all, I think the reader deserves a more even-handed comparison.

Reviewer #2:

None

Reviewer #3:

Remarks to the Author:

Thank you for giving me the opportunity to evaluate the newly revised version of the paper.

My previous concerns were previously addressed.

I have no additional comments or remarks on the new version.

Reviewer #1 (Remarks to the Author):

The authors have made some improvements in the paper. It is a pity that their method does not appear to be able to give any idea of the size of effects as in an elasticity. I note that you find results “ranging from 5% for the carbon tax in British Columbia to 21% for the RGG” These results are quite remarkable and strange. The price level in RGGI is just around say 6 USD while it is more like 65 (Canadian) dollars in BC: if you get a much bigger reaction from a much smaller price signal in RGGI you should ask yourself what is going on. Presumably there are a number of other confounding factors. Naturally producers and consumer react to the sum of all policies and market factors and a complete study should preferably try to account at least for the main ones.

As one example of the text we can take the following “The findings suggest that the individual context and policy design of the schemes best explain the heterogeneity in achieved emissions reductions. These are the most relevant explanatory factors despite controlling for broader policy design features like the sectoral coverage or the design as carbon tax or carbon trading scheme as well as for study design features of the primary studies.”. Well sure – context and design matter – but it seems to me that the size of the carbon price increase is the elephant in the room. Of course it is very likely that a big carbon price will have more effect than a small one....

However given the fact that your method is more focused on proving whether or not there is an effect at all, it is much better to remove numbers that are misleading. Numbers with “size effects” that are based on a mix of studies with widely varying price signals will just be confusing.

I still see however, at least one very substantial hurdle left to deal with before this paper is ready to make the kind of truly complete and important contribution that one would expect from a publication in Nature Communications.

I take the trouble to argue again about these issues because your subject is a very important one and your paper could be seminal and really important but to achieve this, careful consideration and careful writing is required. Let me first of all remind you that you do start your whole paper by saying:

“Countries are not on track to meet the climate goals they committed to under the Paris Agreement [1, 2]. To step up implementation, learning about what policy instruments work in reducing emissions at the necessary speed and scale is critical.

This is a tall order. It is one of humanity’s big challenges and I think we agree that this issue is very important – academically and also for the sake of climate policy. As you know, I worry that you exclude a lot of literature that you don’t see because you are just looking for objects that fit into the framework of an event study and carry the label “carbon price”. I believe that firms and individuals react to the total price of an energy carrier like fuel oil or diesel (and other relevant factors like price of alternatives). If the price is high they use less and if the price is low they use more. The main difference across the world in prices is in fact due to policies like taxes. Sometimes these policies carry other names like gasoline tax, “import duty” or something else instead of carbon tax. However I don’t think that the name of the tax is decisive for use – just its size – or rather the size of the resulting price on the market. I don’t think motorists care what the minister said or if it was a minister rather than a monopoly that charged a high price.

When you speak of fuel elasticity studies you complain that they don’t make a distinction if the price went up through a tax or through other market effects. I do agree there is a difference (and I think some studies have tried to address that although it is difficult) – but I am convinced this is a second order problem compared to not including any price at all or estimating together all kinds of “carbon tax events” whether they imply a big or a small price increase and thus getting no estimate of size.

When it comes to really understanding “what policy instruments work in reducing emissions at the necessary speed and scale”, I must say that I find studies that provide studies with an elasticity estimate much more useful than event studies that just give a yes or a no. I could agree however that there is room for both. Elasticity studies cannot always be readily performed and work better for sectors such as the transport fuels than for economy wide taxes. I can also partly sympathize with your argument that you have already done a lot of work, you are a large team of experienced researchers, you have worked for more than 2 years already ... and basically that it would not be reasonable to ask you to do a whole lot of work of another kind. I am a little divided on this argument because of course work input does not guarantee publication in a good journal. You also have arguments about the size of elasticity estimates – “demand responses to volatile market-induced fuel price changes are much smaller than responses to fuel tax changes”. Well, maybe, However I am again not quite convinced – your own method does not give any size of elasticity estimates at all.

Anyway I am intent on being constructive and do not want to fall in the category of a quarrelsome and one-sided referee. I think that we can agree that there are different methodological approaches. I think an overview of the kind of studies that you provide is useful. I am hoping you will upon reflection, agree that elasticity studies are useful too. Both have their advantages and their weaknesses. I tend to think that identifying the size of the price increase and the response it generates is very important and I think you find other factors to be more important. This divergence is ok and we can continue to reason about it. It is however imperative to give the reader a reasonable and fair description of the two methodologies and their respective advantages and disadvantages.

At the moment you only have a short section starting with the words “Beyond the assessments of explicit carbon pricing policies...” which I find a little ideological and misrepresentative. There are many hundreds of elasticity studies and a very large number of surveys of these studies and they generally agree – within a reasonable margin – that price elasticities are quite high (in the long run). (it is not really my job to give a number but in the past I used to think that -0.7 was a reasonable value, that agreed with averages or meta-analyses of hundreds of studies).

In sum I regret the fact that you do now want to integrate this body of work into your own, I do think that a full scale integration and comparison would make a truly impactful and seminal piece, but I can see that this would be a lot of work and face some considerable challenges. I would therefore be prepared to accept that you don’t do that for the present article. I do however think that this body of work – and this methodological approach is of such importance that it deserves a serious treatment, rather than a mere dismissal. In my opinion the paper needs, at the very least, to be complemented with such a serious description of this literature that says something of its advantages rather than just dismissing it as not scientific, not formal, very small effects etc that I do not find to be a serious or fair summary ... Considering that your method does not give effect sizes at all, I think the reader deserves a more even-handed comparison.

Thank you for the intensive engagement with our study. It has helped us to better reflect on the system boundaries of the study, adequately connect all important strands of literature and make the language more precise. We hope that you appreciate the series of changes that we have implemented across the different round of reviews. We agree with you that there are different relevant strands of the literature to understand the effect of prices on emissions. Let us emphasise that we see the relevance of the fuel price elasticity literature, and you will see that we have connected it now thoroughly to the manuscript. We appreciate your invitation for a constructive solution and have made another genuine effort to respond to this remaining concern that the literature on price elasticities is underrepresented in our study.

Before outlining our amendments to the text, we want to briefly outline the conceptual linkage between the different strands of literature. Our focus here is on carbon pricing policies imposed on the carbon content. With this focus, we link our review to the large theoretical literature, proposing this policy instrument as an efficient measure to increase prices in relation to the emission intensity. As such, carbon pricing can efficiently prioritise the most polluting processes across different fuel types and technologies as price increases are largest where most carbon is emitted. Other price increasing policies are specific for a particular fuel and work only well for efficient mitigation, if the carbon content of the fuel is fairly homogenous – this is the case for petrol, but less so for different coal or oil products. It is this unique feature that has motivated us to focus our study on carbon pricing policies. In general, we are interested in evaluating carbon prices comprehensively – in terms of their introduction effect as well as in terms of carbon price elasticities. However, there are only very few studies on carbon price elasticities so that we focus our quantitative analysis on the assessment of introduction effects, but we later discuss the initial evidence on carbon price elasticities qualitatively.

Fuel price elasticity studies contribute to the question how price increases can reduce emissions from a different angle. This complementary but distinct strand of the literature provides evidence for how the demand for the studied fuel responds to an increase in the price for that fuel. As price changes are frequently observed the data on prices and consumption of fossil fuels provides an excellent environment for quantitative assessments of the marginal effects. The comprehensive literature on price elasticities is commonly conducted for single fuels in the transport or energy sectors. This literature – in contrast to carbon pricing – has been frequently reviewed and meta-analysed. As a response to your comments we connect to this complementary strand of literature now by discussing the conceptual differences and linkages, citing the relevant reviews and meta-analyses of this literature as well as discussing the range of the observed elasticities.

Due to our focus on carbon prices (and the insufficient evidence on carbon price elasticity studies in the literature), we focus our study on the quasi-experimental event studies that use the single event when the carbon pricing policy is implemented. This literature has not been meta-analysed before and traditional, non-quantitative reviews have provided misleading results. While such studies are restricted in that they focus in a single price increase (with variations in the price level only being observed between schemes), they study a policy instrument that is applied across fuels and sectors and applies a uniform price on all fuels based on their carbon contents. For climate policy, this is a very relevant distinction to fuel specific taxes, as mentioned above, as it allows for different reactions in the demand for different fuels. In the power sector, for instance, we regularly observe that natural gas consumption increases with the introduction of a carbon price, due to its comparative advantage over coal, based on their lower carbon contents. The effectiveness of the policy therefore needs to be studied across fuels. The reviewed studies use established methods, like difference-in-differences, synthetic control, or regression discontinuity in time, which are designed to capture the effect of a specific event, here a policy shock.

A third strand of evidence we see very relevant are studies that assess demand responses with respect to changes in the carbon prices. As we highlight in our manuscript as a main conclusion, we see this as a gap in the primary evidence base that should be urgently filled. We also feel that the time is ripe now, as higher carbon prices have been more frequently observed over longer periods of time – like in the European Emissions Trading Scheme. While we particularly searched also for such studies using our comprehensive search strategy, we identified only nine studies assessing the response of emissions to changes in the size of the carbon price. The nine studies provide semi-elasticities of emissions with respect to the carbon price. While we would have been keen to synthesise these studies separately as they provide additional information and could speak to the

questions you are also raising, the evidence base is too small for a formal assessment. In addition, the semi-elasticities provided in the studies are reported in different units and are partly estimated as non-linear relationships. We were thus only able to qualitatively summarise their main findings. We see an important avenue for future research to fill this gap, as this could very much close the gap between the fuel price elasticity literature and the quasi-experimental literature on the policy introduction.

In consideration of your comments and in the desire to iron out any language that could be perceived dismissive, we amended the manuscript to reflect this discussion and carefully worked on our wording to provide the reader a balanced view on the relevant literature. We do not aim to take any position as to what extent price elasticity and policy evaluation studies are more or less relevant for scientific or policy discussions. We discuss in the article now how these strands of the literature can complement each other, while acknowledging that an article in Nature Communications requires a concise presentation and discussion of the findings, which does not allow for a more detailed discussion of the two methods and their respective advantages.

In particular, we respond to your critique by (i) clarifying our research aim and the value of the reviewed literature, (ii) providing a comparison of the reviewed literature with the fuel price elasticity literature, (iii) and highlighting the gap in the literature of studies estimating price elasticities for carbon pricing instruments.

We amended the introduction, to make it very clear from the beginning where we see the contribution of our study. We define the type of studies we are interested in and state our research question. We distinguish how the literature we review can answer the policy relevant question and how this relates to the literature for price elasticities. We state that our meta-analysis fills a gap in the literature and complements the available meta-analyses conducted for the price elasticity studies.

We aim to systematically review the empirical literature on the effectiveness of carbon pricing policies in reducing GHG emissions. While there are other market based policy instruments, such as fuel taxes, import taxes or value added taxes, we focus here on policies which impose a carbon price across fuels based on their carbon contents. One way to assess the effects of carbon pricing is to evaluate experiences in the real world. A growing scientific literature has provided quantitative evaluations of the effects of different carbon pricing schemes on emissions [14, 15, 16]. This evidence is usually provided in the form of quasi-experimental studies which assess the effect of the introduction of the policy (treatment effect). Based on this evidence, our meta-analysis addresses the question: What was the emission reduction effect of the introduction of a carbon price during the early years of its application? This is different from the question, how emissions respond to gradual changes in existing carbon prices. There exist only very few studies estimating this relationship between the carbon price level and emissions [17, 18, 19]. The comprehensive literature on the elasticity of fuel use in response to fuel price changes has been reviewed before in a number of meta-analyses [20, 21, 22, 23, 24].

We focus on the growing evidence base on the effectiveness of introducing a carbon price.

In the discussion we relate our findings to the evidence from price elasticity studies. We summarise the findings from the meta-analyses on fuel price elasticities, explaining the distinction between the strands of evidence, and how that literature can complement the studies we reviewed. We highlight

that the evidence from quasi-experimental studies is unable to provide marginal effects and that the best available source of evidence for this question would be the fuel price elasticity literature.

Even if across schemes the price level of the carbon price, is not found to be the relevant driver of the emissions reductions achieved with the introduction of the policy, within a scheme the effectiveness is expected to increase with increasing prices. This is well studied for other changes in fuel prices, which are found to substantially reduce its consumption [54, 55]. That literature studies all possible price changes on a single fuel, while the here assessed literature on carbon prices studies the effect of a single policy instrument across all fuels. It is thus a complementary but distinct body of evidence. Meta-analyses estimate a reduction of fuel consumption between 0.31% and 0.85% in the long run for a 1% increase in the fuel price [17, 18, 19, 20, 21].

We underline in the discussion section that for implemented carbon taxes and emission trading schemes, price elasticities have not been extensively studied. We summarise the evidence from the nine studies we identified that estimate semi-elasticities for carbon pricing policies. We point out that it would be an important avenue for future research to study the marginal effects of carbon prices. The scarce evidence we have on this question motivates this even further, as it suggests that the response to carbon prices is different from the response to other price changes. We relate this to the comprehensive evidence on fuel price elasticities and call for future research to investigate this further.

Within the literature evaluating the policy effectiveness we identified only nine primary studies estimating semi-elasticities of carbon prices. Four are using the stepwise introduction of the carbon tax in British Columbia to estimate elasticities for the transport and buildings sectors [14, 61, 62, 63], while one is conducted respectively for RGGI [60] and EU ETS [64]. In addition, some studies estimate elasticities across countries and carbon pricing schemes [65, 66, 15]. These studies support what was already known from studies on the price elasticity of fuel consumption [54, 55, 17, 18, 19, 20, 21]: increasing prices reduce fuel use and emissions. Hence, as carbon prices further rise after the introduction additional emissions reductions are achieved. Interestingly, some studies suggest that an increase in the carbon tax leads to larger emissions reductions than an increase of the same size in the market price of the fuel [14, 61, 62, 63]. It will thus be a relevant avenue for future research to understand whether it is a generalisable finding that price elasticities are higher for policy induced price changes compared to market price changes of fossil fuels. Such research could draw on the comprehensive evidence from the fuel price literature.

We furthermore appreciate your concerns regarding the unexplained differences between the effects in different countries that cannot be explained by the level of the carbon price. Throughout the review process, we have taken this concern very serious and your comments have immensely helped us to improve the article in that respect. We acknowledge that our review does not provide a final answer as to which context or policy factors can explain the differences in the effect sizes across schemes. The comments and questions raised during the review process, from our perspective, helped to outline this more clearly. As our meta-regression method, which uses novel Bayesian averaging methods, is not able to fully resolve this question and attributes a lot of the heterogeneity to differences between countries and schemes, we discuss potentially relevant factors in our

discussion section. We believe much of this can be attributed to different abatement costs across economies together with the policy and market factors you mention. Unfortunately, even our Bayesian averaging method does not allow us to include an unlimited amount of independent variables in our meta-regression. We instead provide a transparent overview of the effect sizes measured for each of the reviewed schemes. These will allow future research to investigate the differences further. In fact, we are ourselves exploring other research methods, like mixed-qualitative-quantitative reviews, that could in future provide a better understanding of which factors are most important to explain these differences. Such methods are not yet readily available and could therefore not be applied as part of this study. We are confident that our presentation of the large and partially unexplained heterogeneity of effect sizes may also be picked up by other researchers with brilliant ideas how to address the research questions raised by it. For the moment we took the chance to further refine our discussion of this finding. The section now reads as follows:

Our heterogeneity analysis does not identify a relationship between the price level and the achieved emissions reductions, i.e. the size of the emissions reductions observed across schemes from the introduction of a carbon price cannot be explained well by the carbon price level. This is not surprising as marginal abatement costs may differ widely as, for example, prominently acknowledged in the literature on linking carbon price schemes [52, 53]. It is further different from the expectation that higher carbon prices lead to larger emissions reductions within a carbon pricing scheme as commonly found in available assessments of fuel price elasticities [54, 55, 21]. In line with this argument, we find that the relationship between carbon price levels and emissions reductions in our meta-analytic framework is dominated by the across-scheme variation in prices, which accounts for 91% of the variation in our dataset while the variation within schemes only accounts for 9%. The interpretation for not finding a clear relationship should thus rather be that when implementing a carbon price in two countries with different country contexts, the country with the higher carbon price would not necessarily experience the higher emissions reductions.

This can be observed, for instance, when looking at the cases of China, the EU, and British Columbia. The reviewed literature finds larger emissions reduction effects for the pilot emission trading schemes in China (-13.1%) than for the EU ETS (-7.3%) and the carbon tax in British Columbia (-5.4%), despite the very low carbon prices of the Chinese schemes. The average prices of the eight Chinese pilot schemes are all below US\$ 8 during the study period, while the average prices for the EU and British Columbia are at US\$ 20 and US\$ 18, respectively. This is likely a result of lower abatement costs in China [56] together with differences in the policy contexts of the countries. The effectiveness is certainly influenced by other policies in place. In China indirect carbon prices are lower than in the EU countries and Canada [57], allowing for a higher marginal effect of the implementation of the ETS pilots in China. Non-pricing instruments also diverge across countries. In addition, the implementation of a carbon price (even with a low price) can have a signalling effect towards the emitters, underlining the commitment of the government towards climate mitigation. Evidence for the Guangdong province suggests that signalling has significantly contributed to the achieved emissions reductions in the context of the introduction of the ETS pilots in China [58]. Another example highlighting the relevance of the context of the policy implementation is the case of the RGGI.

The policy implementation coincides with the shale gas boom, which drastically reduced the prices of natural gas in the USA and started around the same time as the RGGI was implemented. In face of these general price dynamics in the US energy sector, RGGI participating states reduced their emissions considerably stronger compared to non-regulated states [59, 60], while the carbon price was only US\$ 3 on average.

Reviewer #3 (Remarks to the Author):

**Thank you for giving me the opportunity to evaluate the newly revised version of the paper.
My previous concerns were previously addressed.
I have no additional comments or remarks on the new version.**

Thank you for your positive feedback.